

# Miocene high elevation and high relief in the Central Alps

Emilija Krsnik[1,2], Katharina Methner[1,5], Marion Campani[1], Svetlana Botsyun[3], Sebastian G. Mutz[3], Todd A. Ehlers[3], Oliver Kempf[4], Jens Fiebig[2], Fritz Schlunegger[6], Andreas Mulch[1,2]

[1]Senckenberg Biodiversity and Climate Research Centre (SBiK-F), Frankfurt (Main), 60325, Germany
5   [2]Institute of Geosciences, Goethe University Frankfurt, Frankfurt (Main), 60438, Germany
[3]Department of Geosciences, University of Tübingen, 72076 Tübingen, Germany
[4]Federal Office of Topography swisstopo, Geologische Landesaufnahme, Wabern, 3084, Switzerland
[5]Department of Geological Sciences, Stanford University, CA 94305, USA
[6]Institute of Geological Sciences, University of Bern, Bern, 3012, Switzerland

10   *Correspondence to*: Emilija Krsnik (emilija.krsnik@senckenberg.de)

**Abstract.** Reconstructing Oligocene-Miocene paleoelevation contributes to our understanding of the evolutionary history of the European Alps and sheds light on geodynamic and Earth's surface processes involved in the development of Alpine topography. Despite being one of the most intensively explored mountain ranges worldwide, constraints on the elevation history of the European Alps, however, remain scarce. Here we present stable and clumped isotope geochemistry measurements to provide a new paleoelevation estimate for the mid-Miocene (~14.5 Ma) European Central Alps. We apply stable isotope δ-δ paleoaltimetry on near sea level pedogenic carbonate oxygen isotope ($\delta^{18}O$) records from the Northern Alpine Foreland Basin (Swiss Molasse Basin) and high-Alpine phyllosilicate hydrogen isotope ($\delta D$) records from the Simplon Fault Zone (Swiss Alps). We further explore Miocene paleoclimate and paleoenvironmental conditions in the Swiss Molasse Basin through carbonate stable ($\delta^{18}O$, $\delta^{13}C$) and clumped ($\Delta_{47}$) isotope data from three foreland basin sections in different alluvial megafan settings (proximal, mid-fan, and distal). Combined pedogenic carbonate $\delta^{18}O$ values and $\Delta_{47}$ temperatures ($30 \pm 5°C$) yield a near sea level precipitation $\delta^{18}O_w$ value of -5.8 ± 0.2‰ and in conjunction with the high-Alpine phyllosilicate $\delta D$ record suggest that the region surrounding the SFZ attained surface elevations of >4000 m no later than the mid-Miocene. Our near sea level $\delta^{18}O_w$ estimate is supported by paleoclimate (iGCM Echam5-wiso) modeled $\delta^{18}O$ values, which vary between -4.2 and -7.6‰ for the Northern Alpine Foreland Basin.

## 1 Introduction

Past elevations of mountain ranges provide insight into the coupled climatic and geodynamic processes that shape orogenic belts. The topographic evolution of continent-continent collision zones such as the European Alps is mainly controlled by isostatic compensation of crustal and/or lithospheric deformation caused by plate convergence (e.g. Beaumont et al., 1996; Schmid et al., 1996; Willett et al., 1993). Despite being a prime example of continent-continent collision following protracted convergence and subduction of oceanic lithosphere (e.g. Frisch, 1979; Froitzheim et al., 2008; Schmid et al., 1996; Stampfli et al., 1998), recent studies in the Central European Alps suggest that compressional tectonics may not be the sole driver for





surface uplift and thus for the formation of Alpine topography (e.g. Kissling and Schlunegger, 2018; Schlunegger and Kissling, 2015). Convergence between the European and the Adriatic plates commenced in the late Cretaceous and led to the subsequent collision between both continental plates at ca. 35 Ma (Schlunegger and Kissling, 2015). Postcollisional slab breakoff (at ca.

32 Ma), and continuing slab rollback of the subducting lithosphere beneath the Alpine arc associated with lithospheric delamination, may have contributed to the rise of topography (Kissling and Schlunegger, 2018). The European Alps are one of the most intensively investigated mountain belts worldwide, and yet there are surprisingly few studies addressing its surface uplift history.

Previous studies of the Central Alps suggested Oligocene to Miocene paleoelevation estimates ranging from mean elevations

of $1900 \pm 1000$ m (Miocene; Schlunegger and Kissling, 2015), and $2300 \pm 650$ m (early Miocene; Kocsis et al., 2007), to $2850 + 800/-600$ m (middle Miocene; Campani et al., 2012). Summit (or maximum) elevations have previously been estimated at 2500–3000 m (Miocene; Kuhlemann, 2007) to at least 5000 m (middle Miocene; Sharp et al., 2005) and potentially >5000 m (Oligocene/Miocene boundary; Jäger and Hantke, 1983, 1984). These estimates, however, are in contrast to geomorphologic and sediment budget-based modeling studies suggesting that present-day elevations of the Alps were attained only at ca. 5–6

Ma while Miocene topography was still much lower (Hergarten et al., 2010).

In this study, we complement previous work by applying stable isotope ($\delta$-$\delta$) paleoaltimetry (Mulch, 2016) using authigenic soil carbonates from the near sea level Northern Alpine Foreland Basin (NAFB) and contrast these with hydrogen isotope data from syntectonic high-Alpine fault zone silicates from the Simplon Fault Zone (SFZ). Our application of Miocene stable isotope paleoaltimetry data builds upon (and revises) former stable isotope paleoaltimetry estimates (e.g. Campani et al. 2012;

Kocsis et al., 2007) as combined analysis of well-dated mid-Miocene sediments (e.g. Kälin and Kempf, 2009; Kempf and Matter, 1999; Schlunegger et al., 1996) along and across two Miocene Alpine foreland megafans (Napf, Hörnli) with clumped isotope ($\Delta_{47}$) thermometry data permits refining the soil carbonate-based near sea level $\delta^{18}O$ estimate of meteoric water ($\delta^{18}O_w$). $\delta^{18}O_w$ values used in paleoelevation reconstructions can be affected by paleoclimate change (Botsyun et al., 2016, 2019; Botsyun and Ehlers, 2021; Ehlers and Poulsen, 2009; Insel et al., 2012; Mulch, 2016; Poulsen et al., 2010). For instance, the

sensitivity of $\delta^{18}O_w$ to regional, global, and topographic variations in paleotemperature, environmental conditions of an air mass prior to orographic ascent, evapotranspiration, water vapor recycling, and changes in vapor source has been shown to introduce uncertainties in stable isotope based elevation reconstructions (e.g. Mulch, 2016; Botsyun et al., 2020, Botsyun and Ehlers 2021). In particular, isotopic changes over continental Europe could be related to a variety of factors such as: declining $p$CO$_2$ levels (Pagani et al., 1999), variable ocean circulation and sea surface temperatures (Flower and Kennett, 1994; Wright

et al., 1992), sea-level fluctuations (Foster and Rohling, 2013), paleogeographic changes (Herold et al., 2008; Poblete et al., 2021), and to other processes affecting $\delta^{18}O_w$ (Botsyun et al., 2019; Poulsen et al., 2007; Risi et al., 2008; Roe et al., 2016; Sewall and Fricke, 2013; Sturm et al., 2010). We thus compare our newly refined near sea level $\delta^{18}O$ estimate with paleoclimate simulations from the isotope-enabled ECHAM5-wiso atmospheric general circulation model (iGCM) which predicts changes in $\delta^{18}O$ of precipitation.





## 2 Study area

### 2.1 The Alps and the Swiss Molasse Basin

The European Alps formed as a result of the northward drift of the Adriatic microplate and the associated formation of a south-directed subduction zone beneath the Tethys Ocean. The Late Cretaceous to Paleogene closure of the Alpine Tethys led to the collision between the Adria and Europe continental plates (Handy et al., 2010; Schmid et al., 1996; Stampfli et al., 1998). Subsequent post-collisional convergence resulted in overthrusting and stacking of nappe sheets (e.g. Schmid et al., 1996). The NAFB formed due to elastic downwarping of the European lithosphere resulting from subduction slab load and topographic load and accommodated eroded material from the N-ward propagating Alpine thrust front (Fig. 1; e.g. Matter et al., 1980; Schlunegger and Kissling, 2015). The Oligo-Miocene Swiss Molasse Basin (SMB) represents the central part of the NAFB (Fig. 1). Deposition of several km-thick sequences of basin fill in the SMB started in the Early Oligocene (e.g. Pfiffner, 1986; Kempf et al., 1997; Kempf et al., 1999) and continued until late Miocene/early Pliocene when basin inversion led to erosion of Molasse sediments (Cederbom et al., 2004, 2011; Mazurek et al., 2006). During this period deposition changed twice from shallow marine to terrestrial resulting in two regressive shallowing-, coarsening and thickening-upward megacycles. The SMB is, therefore, divided into four lithostratigraphic units: the Lower Marine Molasse ("Untere Meeresmolasse", UMM), the Lower Freshwater Molasse ("Untere Süßwassermolasse", USM), the Upper Marine Molasse ("Obere Meeresmolasse", OMM) and the Upper Freshwater Molasse ("Obere Süßwassermolasse", OSM) (e.g. Matter et al., 1980; Schlunegger et al., 1996). Advancing surface uplift of the Alpine mountain belt led to formation of extended drainage networks and alluvial megafans since ca. 32–30 Ma (Kempf et al., 1999; Kuhlemann and Kempf, 2002; Schlunegger et al., 1997) that formed large dispersal systems with cross-sectional widths of nearly 30 km and corresponding stream lengths of >150 km (Schlunegger and Kissling, 2015). A reduction in sediment flux (Kuhlemann et al., 2001) paired with ongoing basin subsidence resulted in a shift from basin overfill (Lower Freshwater Molasse) to underfill (Upper Marine Molasse) at 20 Ma (Garefalakis and Schlunegger, 2019). A subsequent increase in erosional flux (Kuhlemann et al., 2001) together with a lowering of the eustatic sea level (Garefalakis and Schlunegger, 2019) enabled propagation of the fan deltas towards the basin center and led to a shift to an overfilled basin again at ca. 17 Ma, and therefore to the establishment of terrestrial OSM sedimentation by that time (e.g. Kuhlemann and Kempf, 2002).

### 2.2 Alluvial megafans of the OSM

The Napf and Hörnli megafans (Fig. 1B), among other fan systems, formed at the Alpine front and merged into a basin axial drainage system in the central part of the SMB. Deposition of the Napf and Hörnli megafans initiated during the OMM (Kuhlemann and Kempf, 2002), or possibly earlier (Garefalakis and Schlunegger, 2019; Schlunegger and Kissling, 2015). Persistent progradation and accumulation of debris formed subaerial deltas and the relief of the SMB megafans may have reached several hundreds of metres above base level depending on distance to the apex and slope geometry (Garefalakis and Schlunegger, 2018). During the youngest depositional phase (OSM; ca. 17–11 Ma) sediments of SMB megafans are





predominantly composed of amalgamated conglomerate and sandstone packages with mudstone interlayers at the basin margin and alternations of finer grained sandstone beds, mudstones, and marls in the basin center. Sediments of the Hörnli megafan document a short-lived marine ingression at ca. 18.5–18.0 Ma (Bolliger et al., 1995; Keller, 1989), whereas deposits of the

center of the Napf megafan show no evidence of marine sedimentation (Schlunegger et al., 1996). However, marine sedimentation between the megafan deltas continued for at least 1 Myr after the transition from marine OMM to terrestrial OSM within fan deltas (Schlunegger et al., 1996), resulting in lateral facies and elevation changes. As such, the sedimentary sections presented here are in close proximity to the retreating Molasse Sea, yet, depending on their fan position may have developed up to several hundred meters above mid-Miocene sea level. Here we present oxygen ($\delta^{18}$O), carbon ($\delta^{13}$C), and

clumped ($\Delta_{47}$) isotope data from three fully terrestrial SMB sections (Fontannen, Jona, and Aabach) to explore the change in environmental conditions in proximal to distal depositional settings in the Miocene Napf and Hörnli megafans (Fig. 1B), and we relate these data to published stable oxygen isotope data in the evolving Alps (Campani et al., 2012) to update the paleoelevation estimate by Campani et al. (2012) and previous authors for that time.

## 2.3 Sampled sections

Our study of Miocene paleosol carbonate takes advantage of numerous pedogenic soil horizons embedded in a detailed geochronologic framework determined from previous studies of paleomagnetostratigraphy (Kälin and Kempf, 2009; Kempf et al., 1997; Kempf and Matter, 1999; Schlunegger et al., 1996), radiometric age data of volcanic ash layers (Gubler et al., 1992; Schmieder et al., 2018), and mammal biostratigraphy (e.g. Bolliger, 1992; Kälin, 1997). The local paleomagnetic sections were correlated to the astronomically-tuned Neogene timescale (ATNTS2012) of Hilgen et al. (2012).


The almost 1000 m-thick Fontannen section is situated in the proximal part of the Napf alluvial megafan and covers an age range of 17.6–13.3 Ma (Fig. 1B). The section is composed of alternating massive conglomerates, sandstones and silty mudstones of the OSM (Fig. 3). The base of the section (0 m to ~100 m) is composed of the Schüpferegg Conglomerate, which represents the terrestrial equivalent of the marine Luzern and St. Gallen Formations of the OMM (Garefalakis and Schlunegger,

2019; Keller, 1989). The Schüpferegg Conglomerate is overlain by the Napf Beds, which mainly comprise conglomerates and silty mudstones (Schlunegger et al., 1996). Massive conglomerates of up to 100 m thickness dominate the stratigraphy in this region and reveal the proximal position of the section within the alluvial fan (Fig. 2A, 3). Well-developed paleosols with carbonate nodules occur in mud-/siltstone interlayered between the conglomerate beds. Pedogenic horizons are up to 50 cm thick and occasionally show mottling in grey, purple, and yellow. Age constraints of the Fontannen section are given by

deposits associated with the Ries meteorite impact (14.81 ± 0.02 Ma; Schmieder et al., 2018) and by two mammal fauna zones (MN 5/6 and MN 5; Kälin, 1997) .

The 750 m-thick Jona section covers an age range of 16.8–13.7 Ma and is located in the mid-fan to proximal part of the Hörnli alluvial megafan (Fig. 1B). It is mainly composed of alternating conglomerates and mudstones (Kälin and Kempf, 2009).





Similar to the Fontannen section, the stratigraphy of the Jona section is dominated by frequent conglomerate horizons. Almost all outcrops reveal up to several tens of m-thick conglomerates overlaying grey sandstones and marls, the latter characterized by pedogenic overprint (Fig. 2B, 3). Pedogenic features of the Jona section include carbonate nodules, calcified roots, and occasionally strong mottling with grey, yellow and purple colors. In contrast to the proximal Fontannen section, paleosol horizons in the Jona section formed more frequently and are thicker. The up to 2 m-thick paleosols are well developed and

contain abundant carbonate nodules. The section is dated through a projection of the Küsnacht bentonite (14.91 ± 0.09 Ma) to a level situated at ~710 m of the Jona section and through seven mammal sites comprising faunal zones MN 4b, MN 5 and MN 6 (Bolliger, 1992). Furthermore, the Hüllistein conglomerate at ~500 m (16 Ma; Kempf et al., 1997) allows a lateral correlation with the Meilen limestone in the Aabach section 20 km further to the west (Kempf and Matter, 1999).

The 352 m-thick Aabach section covers an age range of 17.3–14.8 Ma (Kempf and Matter, 1999). These sediments were deposited in the distal part of the Hörnli alluvial megafan (Fig. 1B; Kälin and Kempf, 2009) and are characterised by thick mudstones, reflecting typical overbank deposits on floodplains. The lithostratigraphic units consist of alternating sequences of mudstones, marls, siltstones, and fine-grained sandstones (Fig. 3). The typical marl in this section is greyish or yellowish and shows strong evidence of pedogenic overprinting, including frequent mottling in intense purple and yellow colors, root traces

and hackly structures resulting from bioturbation and/or shrink-swell features (slick-and-slide structures) resulting from seasonal wetting and drying (Fig. 2C). Pedogenic horizons often contain abundant carbonate nodules or show caliche formation. The Aabach section reveals the highest frequency of paleosol occurrence of all three SMB sections with pedogenic overprint in almost all mudstones and marls. Furthermore, the characteristic lack of conglomerates and the presence of lacustrine marls and the Meilen limestone at ~170 m indicate that this section was deposited on a floodplain of the distal part

of the alluvial megafan. Age constraints rely on radiometric ages of the projected Küsnacht (14.91 ± 0.09 Ma) and Urdorf bentonites (15.27 ± 0.12 Ma), and five mammal sites (all in mammal faunal zone MN 5; Bolliger, 1992).

## 3 Proxies and methods

### 3.1 Sampling strategy

Miocene SMB (Swiss Molasse Basin) paleosols show typical characteristics of pedogenic overprinting in former B-horizons

including mottling, root traces and different stages of soil carbonate development (Fig. 2). For all analysed sections we targeted well-developed undisturbed paleosols with micritic pedogenic carbonate nodules typically 1–5 cm in diameter (Fig. 2). In total, we collected 383 pedogenic carbonate nodules from 140 layers of the three sections in different alluvial megafan settings (proximal, mid-fan, and distal) for analysis of $\delta^{18}O$, $\delta^{13}C$ and $\Delta_{47}$. Samples were collected along small rivers and from well exposed sections of the SMB alluvial deposits. We compiled coherent age models for all three sections with a resolution of

≤100 kyr and an error of ± 80 kyr–150 kyr for individual paleosol horizons based on paleomagnetostratigraphy studies (Kälin and Kempf, 2009; Kempf et al., 1997; Kempf and Matter, 1999; Schlunegger et al., 1996). Errors on the age models are based



on uncertainties of sample placement within the stratigraphic section and were calculated for the period of lowest sedimentation rate, thus representing the maximum error. Details on the determined age model and error calculation are given in the Supplementary Material.

## 3.2 Carbonate stable isotope ($\delta^{18}O$, $\delta^{13}C$) analyses and $\Delta_{47}$ paleothermometry

$\delta^{18}O$ and $\delta^{13}C$ data were obtained from pedogenic carbonate nodules in well-developed paleosol horizons. Pedogenic carbonate nodules typically form in soils of arid to sub-humid zones due to chemical precipitation from supersaturated soil water (Cerling, 1984; Cerling and Quade, 1993). Their $\delta^{18}O$ and $\delta^{13}C$ values are controlled by $\delta^{18}O$ of meteoric (soil) water and soil $CO_2$, respectively, and are sensitive to changes in temperature, water availability, soil respiration rates and the proportions of C3:$C_4$

biomass of local vegetation (Cerling, 1984). Carbonate clumped isotope ($\Delta_{47}$) paleothermometry is based on the measurement of the abundance of the rare isotopes $^{13}C$ and $^{18}O$ in the carbonate mineral lattice (Eiler, 2007; Ghosh et al., 2006). The effect of "clumping" of rare isotopes is temperature-dependent and unrelated to the $\delta^{18}O$ value of the water from which the carbonate formed. Pedogenic carbonates have shown to reliably record primary $\Delta_{47}$ values in the SMB sections (Methner et al., 2020). We, therefore, determined $\Delta_{47}$ paleotemperatures for each of the three investigated sections (n=5 samples for Fontannen, Jona,

and Aabach) to calculate $\delta^{18}O_w$ values of meteoric waters. Both, clumped isotope ($\Delta_{47}$), and $\delta^{18}O$ and $\delta^{13}C$ analyses were performed at the Joint Goethe University-Senckenberg BiK-F Stable Isotope Facility (Frankfurt, Germany). Analytical details can be found in the Supplementary Material.

### 3.3 Stable isotope paleoaltimetry

Stable isotope paleoaltimetry relies on the systematic decrease of $\delta^{18}O$ and $\delta D$ values of meteoric water with increasing

elevation on the windward side of an orographic barrier (e.g. Currie et al., 2005; Poage and Chamberlain, 2001; Rowley and Garzione, 2007; Siegenthaler and Oeschger, 1980). With increasing altitude ascending air masses undergo adiabatic cooling and rain out which leads to fractionation-driven depletion of $^{18}O$ in the residual water vapour. With progressive rainout the rainfall becomes increasingly depleted in $^{18}O$. Even though lapse rates of $\delta^{18}O$ and $\delta D$ in precipitation may not necessarily be constant through time and space (Botsyun et al., 2020; Ehlers and Poulsen, 2009; Poulsen et al., 2010), averaged global oxygen

isotope lapse rates show a systematic decrease in $\delta^{18}O$ of meteoric water with increasing elevation (e.g. Poage and Chamberlain, 2001). The present-day Alpine $\delta^{18}O$ lapse rate averages -0.20‰/ 100 m (Campani et al., 2012). The comparison of the $\delta^{18}O$ and/or $\delta D$ values of meteoric waters from age-equivalent low-elevation and high-elevation sites ($\delta$-$\delta$) strengthens paleoelevation reconstructions by reducing the impact of long-term climate change on the isotope proxy records (Mulch, 2016; Mulch and Chamberlain, 2018). In this study we evaluate different low-elevation $\delta^{18}O$ records from foreland basin paleosols

and compare them with an age-equivalent high-elevation $\delta D$ record from the Simplon Fault Zone (SFZ) published in Campani et al. (2012). We then relate the difference in $\delta^{18}O$ of meteoric water between these sites to the difference in paleoelevation and update the previous interpretation presented by Campani et al. (2012) (Fig. 4). Normal fault and detachment systems that



served as pathways for meteoric water, which percolated along such fault zones (e.g. Mulch and Chamberlain, 2007) represent valuable archives for high elevation rainfall $\delta^{18}O$ (or $\delta D$) values. Hydrous silicate minerals that formed synkinematically during deformation within these fault systems undergo isotopic exchange with meteoric-derived fluids. The $\delta D$ values of such synkinematic hydrous minerals hence record the $\delta D$ of the infiltrating meteoric waters and can therefore be used to reconstruct $\delta D$ (and $\delta^{18}O$) of high-elevation precipitation. High-elevation samples used here include hydrous phyllosilicates (muscovite, biotite, and chlorite) from the Zwischbergen segment of the SFZ, a ca. 14.5 Ma major extensional detachment in the Central Alps (Fig. 1) that has been shown to be a conduit for meteoric water (e.g. Campani et al., 2012). The formation of the SFZ is associated with Miocene orogen-parallel extension and coeval with orogen-perpendicular shortening (Mancel and Merle, 1987; Mancktelow, 1992). Crustal extension and development of this major extensional fault system promoted large-scale exhumation of the Lepontine metamorphic dome, which underwent rapid exhumation since 20 Ma. Slip movement along the SFZ persisted from 30 Ma to 3 Ma and peaked between 18 Ma and 15 Ma accompanied by rapid exhumation rates, accelerated footwall cooling with highest rates around 20 Ma (Campani et al., 2010; Grasemann and Mancktelow, 1993; Mancktelow, 1992) and infiltration of meteoric fluids with $\delta D$ and $\delta^{18}O$ values of -107 ± 2‰ and -14.6 ± 0.3‰, respectively (Campani et al., 2012).

Pedogenic carbonates from the SMB sections serve as proxies for near sea level precipitation $\delta^{18}O$ values. The $\delta^{18}O$ values of pedogenic carbonate ($\delta^{18}O_c$) are determined by the $\delta^{18}O$ value of soil water from which the carbonate formed, which in turn is closely linked to local meteoric water (Cerling and Quade, 1993)(Cerling and Quade, 1993). In combination with clumped isotope ($\Delta_{47}$) carbonate formation temperatures we calculate $\delta^{18}O_w$ values of meteoric water based on the $\delta^{18}O_c$ values assuming oxygen isotope equilibrium fractionation (Kim and O'Neil, 1997; updated by Kim et al., 2007). Thus, by linking the high elevation SFZ $\delta^{18}O$ estimate with our temperature-corrected near sea level $\delta^{18}O_w$ value we arrive at an updated estimation of the mid-Miocene elevation for the region surrounding the SFZ and the headwaters of the Napf megafan.

Soils can be affected by evaporative $^{18}O$-enrichment in soil water caused by preferential loss of $^{16}O$, which can result in a bias of the soil water $\delta^{18}O$ values (Cerling, 1984; Quade et al., 2007). Therefore, when calculating low-elevation (near sea level) precipitation $\delta^{18}O_w$ values from the $\delta^{18}O$ carbonate record we rely on the first quartile (lowest 25%) mean $\delta^{18}O_c$ values to avoid any potential synkinematically bias to increased near sea level $\delta^{18}O_w$ values and a resulting overestimation in our paleoelevation reconstructions. We calculate $\delta^{18}O_w$ values for meteoric water for each SMB record for the 15.5 to 14.0 Ma time interval (instead of the entire records) to provide an age-equivalent estimate when compared to the high elevation SFZ data.

### 3.4 Paleoclimate simulations

In particular for regions of variable topography the application of isotope tracking climate models allows us to estimate the impact of global paleoclimatic changes and the influence of surface uplift on regional $\delta^{18}O_w$ values (e.g. Botsyun et al., 2019; Ehlers and Poulsen, 2009; Feng et al., 2013; Poulsen et al., 2010). Here we discuss the results from high-resolution isotope-
enabled ECHAM5-wiso GCM experiments previously described and validated in Botsyun et al. (2020). Different topographic

scenarios (topography experiments) were evaluated in order to quantify the signal of surface uplift preserved in $\delta^{18}O_w$ over the

Northern Alpine Foreland Basin and the Alpine orogen. In the absence of model runs with Miocene boundary conditions we

rely on a pre-industrial model setup (details on the experimental setup and boundary conditions can be found in Mutz et al.

(2018) and Botsyun et al. (2020)). We conducted experiments for four cases with Alpine topography ranging between 150%

and 0% (250m) of the modern mean elevation in 50% increments (Alps150, Alps100, Alps50, and NoAlps).

We evaluate $\delta^{18}O_w$ in three regions: 1) NAFB (low-elevation region), 2) Central Alps (high-elevation region), and 3) near

shore (upwind) area of modern France (distant region), which lies on the preferential moisture trajectory to the northern Central

Alps and the NAFB (Fig. 5A). The last was chosen following Botsyun et al. (2020), who suggested the examination of low-

elevation regions that lie far enough from the high elevation data sites enhance assessment of paleoclimate changes. Results

are presented for summer months (June-July-August (JJA)) because pedogenic carbonate preferentially forms during the warm

season (e.g. Breecker et al., 2009).

## 4 Results

### 4.1 SMB near sea level paleosol carbonate $\delta^{18}O$ and $\Delta_{47}$ data

Carbonate $\delta^{18}O$ data in Fig. 3 is shown as mean value per horizon (1-8 measured pedogenic nodules per carbonate bearing

layer (Fig. SI1, SI2 in suppl. Material for $\delta^{13}C$ data). On an overall scale both, carbonate $\delta^{18}O$ and $\delta^{13}C$ values show no

systematic trend throughout the investigated time interval (~17.5 Ma to ~13.5 Ma), yet with changing variability within each

section. Over the entire sections $\delta^{18}O_c$ values vary between 19.0 and 26.7‰ and $\delta^{13}C$ values between -8.5 and 1.2‰ covering

ranges of 7.7‰ and 9.7‰, respectively. We observe differences in $\delta^{18}O_c$ and $\delta^{13}C$ values among the three foreland basin

records, which we relate to their depositional setting (proximal vs. distal) within the megafans and varying impact of soil water

evaporation and soil productivity/interaction with atmospheric $CO_2$ for $\delta^{18}O_c$ and $\delta^{13}C$, respectively.

Fontannen $\delta^{18}O_c$ values show a tight range and vary between 19.0 and 21.3‰ with a mean of 19.8 ± 0.4‰ (n of individual

nodules = 47). $\delta^{13}C$ values show a higher variability and range from -6.0 to -0.6‰ with a mean of -4.0 ± 1.1‰. $\Delta_{47}$ temperatures

average at 36 ± 7°C (propagated error of the two individual samples) for two pedogenic carbonate samples (MC965b; MC961;

Methner et al. 2020) covering 14.5 Ma–14.9 Ma. Reconstructed $\delta^{18}O_w$ values for meteoric water range from -7.1 to -4.7‰

with a first quartile mean $\delta^{18}O_w$ value of -6.5 ± 0.0‰.

Jona $\delta^{18}O_c$ values range from 20.9 to 26.7‰, with a mean of 22.9 ± 1.5‰ (n = 100). $\delta^{13}C$ values range from -8.5 to 1.2‰ with

a mean of -3.8 ± 2.1‰. The mid-fan Jona section shows a wide range of $\delta^{18}O$ and $\delta^{13}C$ values that cover 5.8‰ ($\delta^{18}O_c$) and

9.7‰ ($\delta^{13}C$). $\Delta_{47}$ derived carbonate formation temperatures were obtained for two pedogenic carbonate samples (18EK192b;





18EK172a) at 14.5 and 14.9 Ma and yield an average temperature of $30 \pm 5°C$. Calculated $\delta^{18}O_w$ values range from -6.3 to -0.7‰ with a first quartile mean $\delta^{18}O_w$ value of $-5.8 \pm 0.2$‰.

Compared to the Jona and Fontannen sections pedogenic carbonates from the distal Aabach section yield the highest $\delta^{18}O_c$
values ranging from 23.0 to 25.9‰ with an average of $24.4 \pm 0.7$‰ (n = 57). $\delta^{13}C$ values vary between -6.2 and -0.1‰ with an average of $-2.3 \pm 1.7$‰. We obtained a $\Delta_{47}$ temperature of $33 \pm 2°C$ (12EK096a) at 15.6 Ma. Consequently, the reconstructed $\delta^{18}O_w$ values range between -3.8 and 0.0‰ with a first quartile mean $\delta^{18}O_w$ value of $-3.1 \pm 0.3$‰.

### 4.2 Paleoclimate simulation of $\delta^{18}O_w$ data

Simulated NAFB (low-elevation) $\delta^{18}O_w$ values for the topography experiments range from -4.2 to -5.0‰ for the NoAlps
experiment, from -4.4 to -5.3‰ (Alps50), from -5.1 to -6.5‰ (Alps100), and from -5.5 to -7.6‰ for the Alps150 experiments (Fig. 5B). For all topography experiments $\delta^{18}O_w$ values decrease with increasing elevation. Simulated Central Alpine (high-elevation) $\delta^{18}O_w$ values range from -4.0 to -4.9‰ for NoAlps, from -4.7 to -6.1 (Alps50), from -6.0 to -9.4‰ (Alps100), and from -7.6 to -12.0‰ for the Alps150 experiment (Fig. 5B). For the distant region all topography simulations show slightly higher $\delta^{18}O_w$ values when compared to the NAFB (low-elevation) setting and range from -3.6 to -5.0‰.

## 5 Discussion

### 5.1 SMB near sea level paleosol carbonate $\delta^{18}O$ record

$\delta^{18}O_c$ values of pedogenic carbonate reflect the $\delta^{18}O_w$ values of soil water and prevalent soil temperatures during carbonate formation. Soil water $\delta^{18}O_w$ values typically decrease with elevation but may be biased to higher $\delta^{18}O_w$ values in response to enhanced soil water evaporation (e.g. Quade et al., 2007). Disentangling the impacts of temperature, elevation, and (soil water) evaporation on the $\delta^{18}O_c$ values of soil carbonates is essential for reconstructing a reliable near sea level $\delta^{18}O_w$ record.

The SMB foreland $\delta^{18}O_c$ records show significant differences along the slope of the SMB megafans with the lowest mean $\delta^{18}O_c$ value at the proximal fan position ($19.8 \pm 0.4$‰ at Fontannen), an intermediate value at the mid-fan position ($22.9 \pm 1.5$‰ at Jona) and highest at the distal fan position ($24.4 \pm 0.7$‰ at Aabach; Fig. 3). Environmental differences between the fan positions are not surprising as the Miocene sediments were deposited in large alluvial megafans which
prograded from the Alpine front towards the basin center covering a present-day downstream distance of ca. 30 km (Garefalakis and Schlunegger, 2019). Accounting for post-Miocene crustal shortening, we consider this a minimum estimate for the Miocene length of the Alpine megafans. These megafans were fed by large drainage networks that led to high sediment discharge and aggradation of several hundred meters of detrital material near the fan apex resulting in an elevation gradient towards the orogenic front. Based on sedimentological data of Late Oligocene deposits of the Rigi megafan (30–25 Ma) in
Central Switzerland, fan surface slopes ranged between approximately 0.2° at the base and 0.9° at the top (Garefalakis and





Schlunegger, 2018). Using a time-averaged fan surface slope of $0.6 \pm 0.2°$ and a fan length of 30 km the proximal part of such a fan would be at elevations of more than $300 \pm 100$ m above the Miocene sea level. Because of this internal elevation gradient, foreland basin records from proximal locations (e.g. Fontannen; Campani et al., 2012) were deposited at higher elevations compared to sections from more distal fan sites, thus not representing near sea level $\delta^{18}O_w$ values.

The distal Aabach section was hence closest to the basin axis (mainly also because of the occurrence of paleolakes that were established in the lowest area of a basin, e.g. Platt and Keller, 1992) which was near or at sea level at that time (Kuhlemann and Kempf, 2002). The $\delta^{18}O_c$ and $\delta^{13}C$ values at Aabach, however, show a strong positive correlation (Fig. SI3 in suppl. Material), and they are additionally associated with high $\delta^{13}C$ values. Because significant $C_4$ vegetation was absent at that time (Cerling and Quade, 1993; Tipple and Pagani, 2007), these values indicate low soil respiration rates during the formation of

pedogenic carbonate (Quade et al., 2007 and references therein). High $\delta^{13}C$ values of soil carbonate are common during high plant water stress, low soil productivity, and low soil respiration and high soil evaporation rates (e.g. Breecker et al., 2009; Caves et al., 2016). Furthermore, the high $\delta^{13}C$ values at Aabach are consistent with a shift from humid to warm and semiarid conditions after 25 Ma, inferred from paleofloral records for the SMB (Berger, 1992). Evaporative effects caused by local aridity and enhanced soil water evaporation may shift carbonate $\delta^{18}O_c$ values to higher values (Cerling and Quade, 1993).

These processes likely account for the strong covariance of the $\delta^{18}O_c$ and $\delta^{13}C$ values (Fig. SI3 in suppl. Material). Consequently, we consider the rather high and highly variable Aabach pedogenic carbonate $\delta^{18}O_c$ values to be biased by varying soil water evaporation that translates into a positive bias in reconstructed soil water $\delta^{18}O_w$ values.

For our paleoelevation reconstruction, we thus consider the mid-fan Jona section as a conservative estimate for near sea level $\delta^{18}O_w$ values as a) its fan position places the Jona section clearly down-slope from the proximal Fontannen section, b) combined

$\delta^{18}O_c$ and $\delta^{13}C$ values show no indication for enhanced soil water evaporation, and c) using the mid-fan Jona section underestimates rather than overestimates reconstructed paleoelevations. Consequently, we consider the variability in $\delta^{18}O_w$ values of the Jona section to reflect secular changes in environmental conditions. To further reduce potential (soil) evaporation bias we only select the first quartile mean $\delta^{18}O_c$ values ($\delta^{18}O_{c, \text{lowest } 1/4} = 21.4 \pm 0.2$‰ compared to mean $\delta^{18}O_c = 22.7 \pm 1.4$‰) to establish our best estimate for a mid-Miocene near sea level $\delta^{18}O_w$ value.

**5.2 Near sea level precipitation $\delta^{18}O$ estimate**

We conducted clumped isotope ($\Delta_{47}$) thermometry on pedogenic carbonates from all three SMB sections in order to obtain estimates of carbonate formation temperatures and $\delta^{18}O_w$ values of meteoric water. $\Delta_{47}$ temperatures show high reproducibility and attain values of 36°C (Fontannen; n= 2 samples), 30°C (Jona; n= 2 samples), and 33°C (Aabach; n= 1 sample), consistent with warm season pedogenic carbonate formation during the warm temperatures of the mid-Miocene Climatic Optimum

(Methner et al., 2020). Using $21.4 \pm 0.2$‰ (1st quartile mean Jona section) as a near sea level carbonate $\delta^{18}O_c$ value, this results in $\delta^{18}O_w = -5.8 \pm 0.2$‰ as our best estimate for Miocene near sea level $\delta^{18}O_w$ values of meteoric water. Collectively, these data revise previous estimates of near sea level $\delta^{18}O_w$ based on the proximal Fontannen section ($\delta^{18}O_w = -8.9 \pm 0.5$ for 21°C;





Campani et al., 2012) which were calculated with mean annual temperatures derived from paleobotanical analysis. The difference to the revised value is composed of a more suitable choice of the near sea level record (mid-fan vs. proximal fan

position) and measured (instead of estimated) carbonate formation temperatures contributing to +1.2‰ and +1.9‰, respectively.

The reconstructed near sea level Jona $\delta^{18}O_w$ value is consistent with $\delta^{18}O_w$ values from mid-Miocene SMB volcanic ash horizons (Bauer et al., 2016). Using a mineral-water fractionation temperature of 30°C (measured $\Delta_{47}$ temperature for Jona) these ash layers (Ries, Küsnacht, and Urdorf) reveal mean $\delta^{18}O_w$ values between -6.1 and -2.9‰, which are equal or even

higher than our conservative near sea level estimate of $\delta^{18}O_w$ = -5.8 ± 0.2‰ based on SMB Jona pedogenic carbonate.

**5.3 Mid-Miocene stable isotope paleoaltimetry of the Central Alps**

We contrast the near sea level ~14 Ma old $\delta^{18}O_w$ record from the Jona section with high-elevation $\delta^{18}O_w$ record from the Simplon Fault Zone at 14.5 Ma (Fig. 4).

Relative differences in $\delta^{18}O_w$ between near sea level and the high-Alpine SFZ fault zone are expressed by following Eq. (1):

$$\Delta(\delta^{18}O_w) = \delta^{18}O_w \text{ (SFZ)} - \delta^{18}O_w \text{ (SMB)} \qquad (1)$$

Given $\delta^{18}O_w$ values for near sea elevation of -5.8 ± 0.2‰ and the high-Alpine SFZ of -14.6 ± 0.3‰ (Campani et al., 2012), respectively, $\Delta(\delta^{18}O_w)$ equals -8.8 ± 0.5‰ (Fig. 6A). For the paleoelevation calculation, we explored different oxygen isotope lapse rates (Fig. 6B), including the present-day Alpine lapse rate based on long-term meteorological station data (-2.0‰/ km; Campani et al., 2012), the present-day surface water and precipitation based oxygen isotope lapse rate for Europe (-2.1‰/ km; Poage and Chamberlain, 2001), a thermodynamic model based lapse rate that tracks the isotopic composition of water vapour

along precipitation trajectories (Currie et al., 2005; Rowley et al., 2001; Rowley and Garzione, 2007), and the output of the isotope-enabled paleoclimate model ECHAM5-wiso (-2.4‰/ km; Botsyun et al., 2020). Independent of the choice of these lapse rates a $\Delta(\delta^{18}O_w)$ value of -8.8 ± 0.5‰ requires significant orographic rainout and hence elevated topography. For instance, applying the different lapse rates to a $\Delta(\delta^{18}O)$ value of -8.8‰ yields paleoelevations in the range of ~3690 m to

4420 m (Table SI5 in suppl. Material). To keep comparability with the previous stable isotope paleoaltimetry study of Campani et al. (2012), we use the modern Alpine oxygen isotope lapse rate of -2.0 ± 0.4‰/ km. Based on the $\Delta(\delta^{18}O_w)$ value of -8.8 ± 0.5‰ this is consistent with a relative elevation difference between the SMB and the SFZ of $\Delta z$ (m) = 4420 ± 770 m. Differences in atmospheric temperature gradients in the past will inevitably affect oxygen and hydrogen isotope lapse rate and climate change may play a decisive role when trying to assess paleoelevation of ancient mountain ranges. Especially for

warmer periods of the Earth's past it is shown that warm conditions yield shallower lapse rates (Poulsen and Jeffery, 2011; Rowley and Garzione, 2007). All existing data point to warmer conditions in Central Europe during the mid-Miocene Climatic Optimum when compared to today (Böhme, 2003; Methner et al., 2020; Mosbrugger et al., 2005; Pound et al., 2012). The mid-Miocene should logically have been characterized by lower isotopic lapse rates when compared to the present. Thus, application of the present-day Alpine lapse rate most likely underestimates mid-Miocene paleoelevations. Our estimated mid-





Miocene (~14.5 Ma) surface elevation of ~4420 m for the region surrounding the SFZ is in good agreement with proposed
      mid-Miocene minimum elevation of 5000 m given by Sharp et al. (2005) and the >5000 m found for the Oligocene/ Miocene
      boundary by Jäger & Hantke (1983, 1984), and places our estimated paleoelevation in the higher spectrum of derived
      paleoaltimetry estimates hitherto.

### 5.4 Modeled low-elevation $\delta^{18}O_w$ estimates

Collectively, the topography-modulated model experiments indicate the following. First, modeled Northern Alpine Foreland
      Basin $\delta^{18}O_w$ values for the Alps50, Alps100, and Alps150 cases range from -4.4 to -7.6‰, and therefore are in good agreement
      with the reconstructed low-elevation $\delta^{18}O_w$ estimate of -5.8‰ from the Swiss Molasse Basin. Furthermore, the ECHAM5-wiso
      simulations reveal that a $\Delta(\delta^{18}O_w)$ value of -8.8‰ between the high-elevation and the low-elevation records requires a
      mountain range with an elevation of >150% of present-day Alpine mean topography (Fig. 5B, 6B). We therefore interpret the
observed difference in $\delta^{18}O_w$ between the high-elevation and low-elevation records as resulting from a higher-than-modern
      mid-Miocene elevation of the region surrounding the Simplon Fault Zone. This interpretation is supported by the predicted
      differences in the 150% to 100% topographic change explored in the paleoclimate simulations.

### 5.5 High (and highly variable) mid-Miocene Central Alps?

      Various mechanisms may have contributed to the increase of post-collisional Alpine surface elevation. These include
horizontal shortening, thickening of the continental crust, and following isostatic adjustment of the lithosphere (e.g. Pfiffner
      et al., 2002), mantle scale uplift caused by slab breakoff (e.g. Lippitsch, 2003), slab rollback and associated delamination
      within the lithosphere (Kissling and Schlunegger, 2018; Schlunegger and Kissling, 2015) or asthenospheric upwelling (e.g.
      Lyon-Caen and Molnar, 1989). At the same time climate change will impact topography, e.g. through accelerated erosional
      unroofing forcing isostatic rebound of the unloaded lithosphere (Cederbom et al., 2004; Champagnac et al., 2009; Mey et al.,
370   2016).
      The history of Central Alpine topography is considered to have started no later than the Late Oligocene/ Early Miocene after
      slab breakoff of subducted oceanic lithosphere ~32–30 Ma ago (Schlunegger and Kissling, 2015). Increased sediment
      discharge in the Swiss Alps has been documented for several periods (30–28 Ma, 24–21 Ma, and 18–17 Ma) and linked to
      exhumation and surface uplift events and increase in topographic relief (Kuhlemann et al., 2001). A transition from a poorly
dissected, plateau-like orogen to a rugged mountain range with steep relief is proposed for the Alps at ~27 Ma as a possible
      response to slab breakoff (Garefalakis and Schlunegger, 2018), promoting the buildup of high mountain peaks, and establishing
      a N-S oriented drainage network with a drainage divide situated close to the area of inferred slab breakoff (Kuhlemann et al.,
      2001). Can stable isotope paleoaltimetry data inform about the spatial heterogeneity of surface elevation in the Mid Miocene
      Central Alps?
Although the currently observed Alpine mean topography with a present-day average elevation of ca. 2000 m (Kühni and
      Pfiffner, 2001) may not have experienced major changes since the timing of continent-continent collision (Kissling and





Schlunegger, 2018), basin- and orogen-related reorganizations are likely to have affected the topographic evolution on a regional scale since at least the early Miocene. These include exhumation in the Lepontine and Aar regions (Herwegh et al., 2017; Schlunegger and Willett, 1999), hypothesized reversal in slab polarity beneath the Eastern Alps (Kissling et al., 2006;

Lippitsch, 2003), switch in the regional tilt of the SMB and an associated change in the basin-axial discharge direction at ~17 Ma and to a more complex pattern thereafter (Berger et al., 2005; Kuhlemann and Kempf, 2002; Kühni and Pfiffner, 2002), a change in sediment provenance in the SMB deposits (e.g. Anfinson et al., 2020; Von Eynatten et al., 1999), and reorganization of the main drainage divide (Kühni and Pfiffner, 2002; Schlunegger et al., 2001). Four observations point to high spatial topographic heterogeneity in the Mid Miocene Central Alps: (1) The rapid nearly vertical rise of the Aar massif at ~20 Ma is

associated with a rearrangement of the drainage network leading to a shift of the drainage divide towards the uplifted crystalline block (Bernard et al., in press; Kühni and Pfiffner, 2001; Schlunegger et al., 2001), which is made up of basement rocks of the European plate comprising granites and granodiorites (Herwegh et al., 2020) with the lowest erodibility in the Central Alps (Kühni and Pfiffner, 2002). A shift of the drainage divide towards rocks with low erodibility implies a decrease in overall erosion/denudation rates by reducing the erosional potential for several Myr. This potentially resulted in increased surface

uplift rates, and, thus in a local rise of topography (Bernard et al., in press). (2) Associated re-routing of streams by rise of such a crystalline block in the center of an orogen is further consistent with the initiation of an orogen-parallel oriented drainage system similar to the change from an orogen-normal N-S-directed to an orogen-parallel E-W-oriented drainage pattern that occurred during the early Miocene. Such a change will inevitably affect the overall relief structure. We need to add, however, that an orogen-parallel pattern was not yet fully established in the mid-Miocene, as studies show that the Napf megafan was

still receiving deposits from the Aar region at this time (Stutenbecker et al., 2019). (3) Interestingly, at the same time, rapid exhumation of the nearby Lepontine area through slip along the SFZ was associated with a reduction in sediment supply to the SMB (Kuhlemann et al., 2001). Normal fault-induced rapid exhumation of the Lepontine dome footwall, with highest rates at ~20 Ma renders it likely that mean elevation decreased in the Lepontine area in response to rapid tectonic exhumation. (4) The stable isotope paleoaltimetry data from the Simplon Pass region, however, indicate rather high >4000 m coeval surface

elevations only 45 km to the W of the Lepontine Dome.

The co-existence of regions with different elevations on a small spatial scale within the Miocene Central Alps points to a regional landscape characterized by significant topographic complexity that was sensitive to alterations of the local drainage network, and was accompanied by laterally variable exhumation rates. Therefore, on a regional scale the Miocene Alps were most probably characterized by a heterogeneous, and spatially transient topography with high elevations locally exceeding

4000 m. A heterogeneous, non-cylindrical Alpine tectonic structure with corresponding topography has implications e.g. for the spatially variably impact of slab dynamics to Earth surface process by provoking perturbations of surface loads as well as for assumptions made in orogen-scale denudation and landscape development models. Subsequent paleoaltimetry studies will have to focus on changes of surface elevation through space and time. The combination of stable isotope paleoaltimetry, analysis of regional exhumation patterns in conjunction with paleoclimate modelling can contribute to establishing not only

absolute elevations but also to reconstructing past elevation difference along and across the orogen by linking long-wavelength





subsurface with short-wavelength surface processes which affect the Alpine topographic evolution. Further experiments with isotope tracking global atmospheric circulation models that test time-specific boundary conditions of the middle Miocene will represent key elements in verifying observed changes in $\delta^{18}O_w$ and will provide important information on links and feedbacks between surface uplift and associated climate change.

**6 Conclusions**

Our revised stable isotope paleoaltimetry estimates indicate high (>4000 m) paleoelevations for the mid-Miocene (15.5–14.0 Ma) Central Alpine region surrounding the Simplon Fault Zone. This result is based on a) the pedogenic carbonate record best representing the near sea level $\delta^{18}O_w$ values in the Swiss Molasse Basin, b) quantification of carbonate formation temperatures by $\Delta_{47}$ thermometry (rather than assuming those from other terrestrial temperature proxies) and c) ECHAM5-wiso modeled

Northern Alpine Foreland Basin $\delta^{18}O_w$ values that cover the same range as reconstructed near sea level mean $\delta^{18}O_w$ values. We link the Miocene reorganization of the Alpine drainage network, initiated by the uplift of the Aar massif at ~20 Ma, with the proposed establishment of a highly variable topography within the orogen. Based on coupled $\delta^{18}O_c$ values and clumped isotope ($\Delta_{47}$) paleothermometry on SMB pedogenic carbonates we estimate a mean $\delta^{18}O_w$ value for near sea level precipitation of -5.8 ± 0.2‰. Our conservative paleoaltimetry estimation yields a difference in paleoelevation of 4420 ± 770 m between the

SMB and the region surrounding the SFZ.

Data availability
Supplementary data associated with this article can be found in the online version.

Author contributions
EK, KM, AM, and TAE designed the study. KM, AM, MC, SB, and OK helped during field work and with sample collection. EK carried out sample preparation and analyses; SB produced the model data. KM and JF supported the clumped isotope analyses. AM, KM, SB, SGM, TAE, and FS discussed the results and EK, KM and AM prepared the paper with contributions by all co-authors.


Competing interests
The authors declare that they have no conflict of interest.

Acknowledgements
This is a contribution to DFG-SPP 2017 4D-MB. We acknowledge support through DFG ME 4955/1-1 (to KM), DFG MU 2845/6-1 (to AM), DFG EH 329/19-1 (to TAE), DFG MU 4188/1-1 (to SGM), and DFG FI 948/4-1 (to JF). KM further



acknowledges support through the Feodor-Lynen-Fellowship of the Alexander von Humboldt foundation. We thank S. Hofmann (Frankfurt) for laboratory assistance.

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

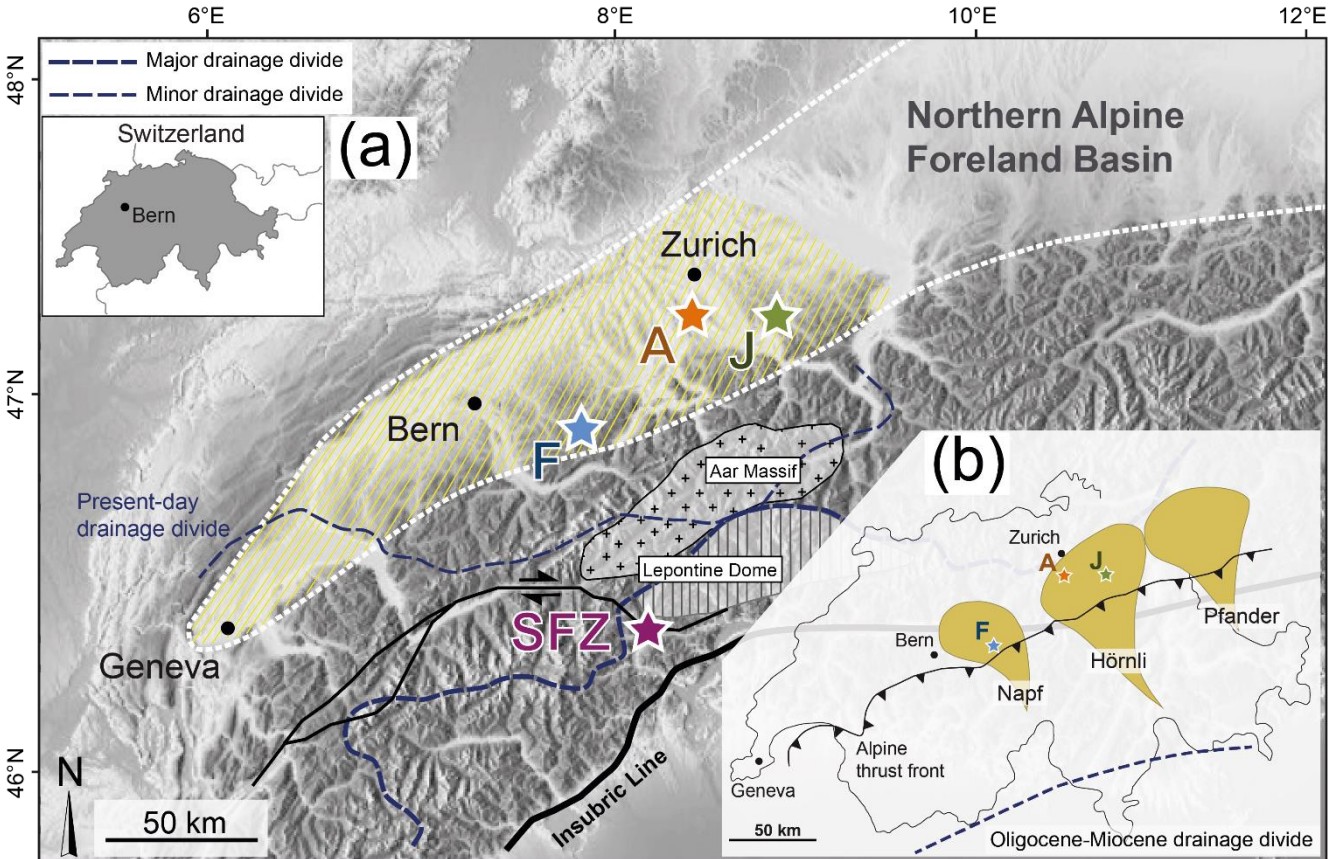

**Figure 1: A) Topographic map of the Central Alps and its northern foreland basin (Swiss Molasse Basin; yellow hatched area) with sections Fontannen (F), Aabach (A), Jona (J) and the alpine Simplon Fault Zone (SFZ) sampling locality. Dashed blue lines mark**
**the present-day drainage divide (after Schlunegger et al., 2007). B) Setting of the foreland basin sections within alluvial megafan deltas during the middle Miocene (15–14 Ma; drawn after Berger et al., 2005). The Fontannen section is located in the proximal part of the Napf megafan, while the Aabach and Jona sections represent a distal, respectively, a mid-fan setting in the Hörnli megafan. Dashed blue line indicates the Oligocene-Miocene drainage divide (after Schlunegger et al., 2007).The logo of Copernicus Publications.**











**Figure 2: Outcrop conditions of the Swiss Molasse Basin sections and close-ups of sampled pedogenic carbonate nodules. A) Proximal Fontannen section: ca. 1 m thick red paleosol profile overlain by massive sandstones and conglomerates (left). Mottled paleosol with carbonate nodules (top right). B) Mid-fan Jona section: ca. 2 m thick paleosol profile with intense mottling intersected by river channel sandstone (left). Ca. 20 cm long calcified root within a paleosol profile (center). C) Distal Aabach section: floodplain deposits with typical strong pedogenic overprint and intense mottling in yellow and red (left). Paleosol abundant in carbonate nodules (right). Also shown are close-ups of individual pedogenic carbonate nodules for each section.**

**Figure 3: Pedogenic carbonate δ¹⁸O values for Swiss Molasse Basin sections. In total, 141 individual carbonate nodules were analyzed for Fontannen, respectively Jona, and 101 for Aabach. Only the mean value for each carbonate bearing horizon is shown here (Fontannen n=50; Jona n=56; Aabach n=34). $\delta^{18}O_c$ values are plotted next to their stratigraphic position. Also shown is the stratigraphic position of the Hüllistein marker bed (H) and its equivalent for distal regions, the Meilen Limestone (M). Each stratigraphic section covers an age range of ca. 17.5 Ma to 14 Ma (see Supplementary Material).**



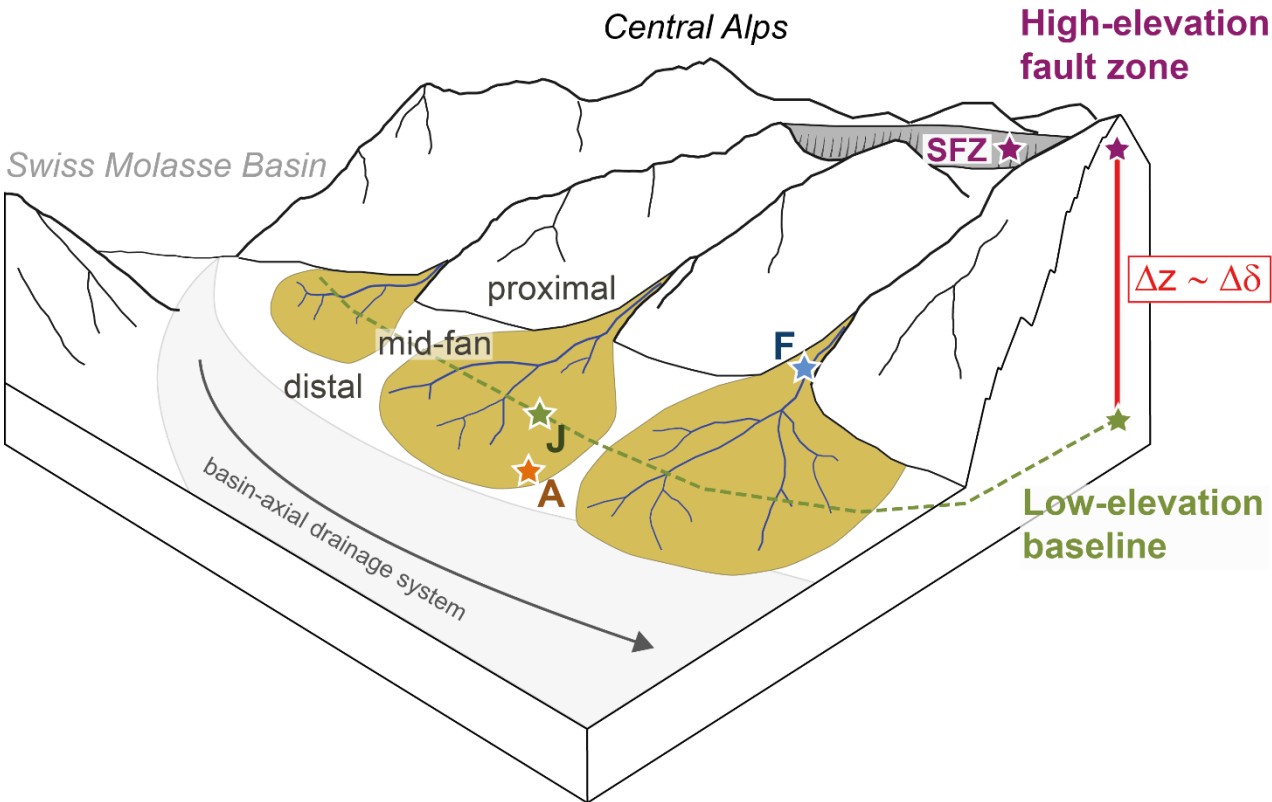

**Figure 4: Simplified sketch of the mid-Miocene Swiss Molasse Basin (SMB, 15–14 Ma) showing the distribution of alluvial megafans along the northern Alpine flank. The sampled sections are located at different paleoaltitudes within the megafan deltas which results in an internal elevation difference of ca. 300 (± 100) m (see text for discussion). We use the difference in precipitation $\delta^{18}O$ values ($\Delta\delta$) between the low-elevation SMB records (F, J, A, see Fig. 1) and the high-elevation (projected) Simplon Fault Zone (SFZ) to calculate the elevation difference ($\Delta z$) between these sites. Grey arrow depicts the paleo-discharge direction of the drainage system. Distribution of alluvial fans adopted from Berger et al. (2005).**



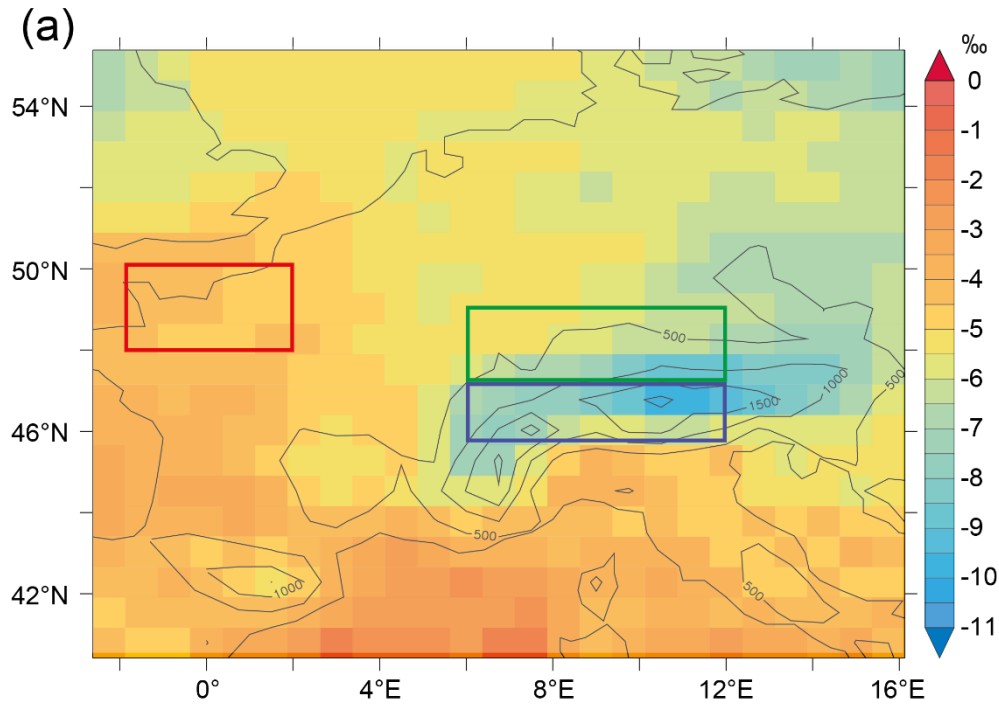

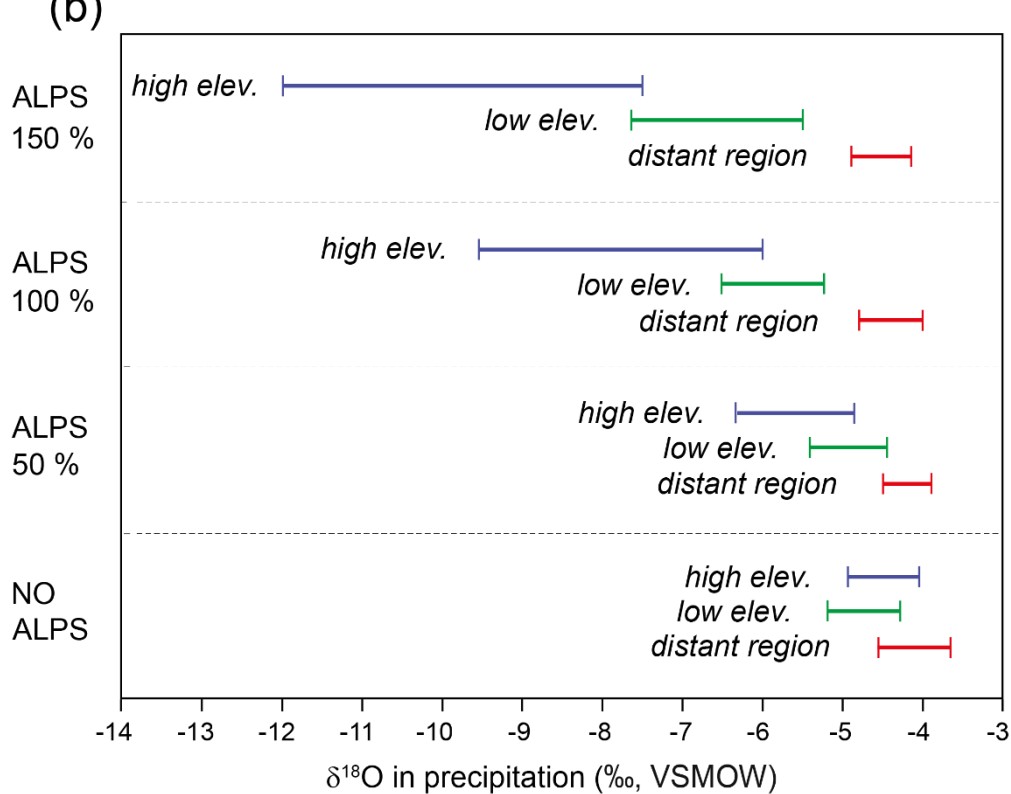



**Figure 5: A)** Spatial distribution of ECHAM5-wiso simulated mean JJA $\delta^{18}O_w$ for the preindustrial (Alps 100%) experiment for three regions: 1) within blue rectangular area for elevations over 1600 (high elev.), 2) within green rectangular area for elevations between 200 m and 600 m (low elev.), and 3) within red rectangular area for distant regions (distant region). **B)** Ranges of JJA $\delta^{18}O_w$ for Alps 150%, Alps 100%, Alps 50%, and No Alps experiments for the selected regions (Botsyun et al., 2020).

**Figure 6: Paleoelevation calculation of the Miocene Central Alps. A)** The difference of precipitation $\delta^{18}O_w$ values ($\Delta(\delta^{18}O)$) between the high-elevation (Simplon Fault Zone, Campani et al., 2012) and the low-elevation sites (Fontannen, Jona, Aabach) is a measure of the respective elevation difference. We use a conservative approach to calculate the paleoelevation and only consider the lowest





25% $\delta^{18}O_w$ values (n=10) of the mid-fan situated Jona section as most suitable to serve as a low-elevation reference baseline (see text for discussion). Swiss Molasse Basin precipitation $\delta^{18}O_w$ values were calculated from pedogenic carbonate $\delta^{18}O_c$ values using carbonate clumped isotope ($\Delta_{47}$) formation temperatures. The figure shows foreland data from this study (Fontannen, Jona and

730    Aabach) and from Campani et al. (2012; Fontannen (2012)). Also shown are quartz vein fluid inclusion $\delta^{18}O_w$ values from the Lukmanier region (Swiss Alps; Sharp et al., 2005) and modeled range for low-elevation precipitation $\delta^{18}O_w$ values for present-day Alps 50%, Alps 100%, and Alps 150% (ECHAM5-wiso iGCM run (Botsyun et al., 2020)). B) Jona $\Delta(\delta^{18}O)$ value and inferred elevation differences ($\Delta z(m)$) (light green rectangle) based on different isotope lapse rates (black line: -2.0‰/ km (Campani et al., 2012); brown line: -2.1‰/ km (Poage & Chamberlain, 2001); yellow line: -2.4‰/ km (Botsyun et al., 2020); grey line: Currie et al.,

735    2005; Rowley & Garzione, 2007). The present-day Alpine $\delta^{18}O_w$ lapse rate of -2.0‰/ km renders the Jona $\Delta(\delta^{18}O)$ value of -8.8‰ to a paleoelevation difference of 4420 m (dashed line in green) between the foreland basin and the intra-Alpine SFZ.