# Peer review of "Miocene high elevation in the Central Alps"

_Solid Earth, 2021_

## Author Comment (AC1)

**Author response:** We would like to thank the referee for the time and effort he dedicated in reviewing our manuscript. We appreciate the referee's insightful comments and suggestions and carefully addressed them. Please see below, in blue and italic font, for a point-by-point response to the reviewer's comments. Provided page numbers refer to the revised manuscript file with tracked changes.

**RC1**: 'Comment on se-2021-59', Jay Quade, 30 Jun 2021 reply

This is an excellent paper that builds upon the foundations of Campani (2012) and Methner (2020) to reconstruct paleoelevation of the Swiss Alps in the mid-Miocene optimum. This review took me extra time because I had to read those papers. This paper fills in the picture by studying low-elevation paleosols from three mid-Miocene sections from the foreland basin.

1) The dating of these looks exceptional, but for the purposes of assessing the diagenesis history, a clear gap on the paper was the lack of discussion on the burial depths and burial temperature history of the molasse basin from other published sources.

- *We would like to thank the reviewer for this insightful comment and agree that this issue could have been stated more prominent in the manuscript. We acknowledge that diagenetic alteration of terrestrial carbonates can be challenging to assess. Initial stable and clumped isotopic compositions of pedogenic carbonate nodules can be altered by diagenetic overprint and burial metamorphism due to increased burial temperatures. In the following we want to discuss why we assume that diagenetic overprinting Molasse Basin's paleosols had no (major) impact on our study:*
    1) *Carbonate-bearing sediments of the Swiss Molasse Basin show primary soil structures (e.g. root traces, mottling, structures from bioturbation and wetting and drying) and remain poorly consolidated lacking signs of diagenetic hardening. This indicates that diagenetic impact on the collected pedogenic carbonate samples has, if at all, remained small.*
    2) *Vitrinite reflectance and apatite fission track data yield maximum erosion estimates ranging from 350 m to 2100 m and maximum burial temperatures of 40°–110°C for Central and Eastern Switzerland. Sedimentation of the youngest basin fill (OSM) ended between ~10 Ma and ~5Ma in the Swiss Molasse Basin. Considering the low carbonate clumped isotope temperatures (30–36°C) and the rather short time interval of max. 10 million years of burial, during which collected carbonate nodules from the OSM were overburden, we can exclude any solid state reordering within the carbonate minerals as the time and temperature are not sufficient (Henkes, G. A., Passey, B. H., Grossman, E. L., Shenton, B. J., Pérez-Huerta, A. and Yancey, T. E.: Temperature limits for preservation of primary calcite clumped isotope paleotemperatures, Geochim. Cosmochim. Acta, 139, 362–382, doi:10.1016/j.gca.2014.04.040, 2014.)*
    3) *Diagenetic overprint should result in rather homogenized $\delta^{18}O$ values, and should shift the $\delta^{18}O$ compositions to lower values (assuming high diagenetic temperatures and $^{18}O$-depleted (meteoric) waters). However, SMB records show high variability in both $\delta^{13}C$ and $\delta^{18}O$ values, which covers 9.7‰ ($\delta^{13}C$), and 5.8‰ ($\delta^{18}O$) for the Jona section. Moreover, a potential lowering of $\delta^{18}O$ would ultimately result in underestimating rather than overestimating inferred paleoelevations.*

- *However, we agree with the reviewer that the paper would benefit from a more detailed assessment of diagenetic impact on Swiss Molasse Basin carbonates. We have therefore included a new paragraph (see lines 327–332 in revised manuscript) and add more details in the Supplementary Material (SI4) addressing estimated amounts of erosion, inferred burial temperatures and potential diagenetic impact on the collected samples.*

2) If this were my paper I would add depth of soil carbonate nodules below the top of the paleosol. Depth has a big influence on d13C, d18O, and D47 values. In the general the paper does not record soil carbonate distribution with depth, and how this varies among soils. But they took so many samples that I assume that that got a representative suite from shallow to deep, which serves the purpose of the paper well enough

- *We very much appreciate this comment and agree with the reviewer that variations of $\delta^{13}C$ and $\delta^{18}O$ with increasing sampling depth have to be taken into consideration.*
- *Based on our field observations we found that carbonate-bearing horizons in the Swiss Molasse Basin occur mainly in sequences of stacked paleosols with eroded A-horizons. In this setting the top horizons of sampled paleosols were sharply truncated by erosional surfaces, which makes it difficult for us to state precise sampling depths for the collected carbonate nodules. Nevertheless, the absence of the upper (A-) horizon in stacked paleosols gives some constraints on the sampling depth and led us to assume that sampling of collected carbonate nodules was performed from below critical near-surface horizons (uppermost ~20–30 cm).*
- *Furthermore, where possible, we tried to identify individual paleosol profiles for the critical time interval of paleoelevation reconstruction (15.5–14.0 Ma) in order to assess depth-related variations of soil carbonate $\delta^{13}C$ and $\delta^{18}O$ values. For these individual profiles (e.g. "Jona" 18EK199-18EK203 (horizons with carbonate nodules: 577.0 m - 577.4 m), 18EK165-18EK177 (488.7 m - 487.6 m)), both $\delta^{13}C$ and $\delta^{18}O$ values show no strong correlation with the sampling depth, which led us to assume that carbonate samples have not been influenced by near surface-related biases of isotopic compositions.*
- *We clarified this in the manuscript and point out that sampled carbonate nodules were mainly collected from former B-horizons (see lines 191–193 in revised manuscript).*

3) Some things that struck me about the results, which might be expanded upon in the revised version, was how high the d13C values of the soil carbonates, averaging -2 to -3‰±1‰. In the absence of C4 plants and high pCO2 (neither are indicated for this period), this indicates fairly modest respiration rates and dry desert conditions of formation, typical of sagebrush covered steppe or drier in the Great Basin (see Quade et al 1989 GSA Bull. Systematic variations…). I wonder: are there other indicators of such aridity in the molasse basin of the mid-Miocene, such as evaporites? I am surprised it was as dry as the Great Basin, given the region is so wet today. Perhaps this reflects some strong rainshadow effects, although I would have thought storms came out of the west, then as now. I find this really intriguing.

- *We fully agree with the reviewer that Swiss Molasse Basin pedogenic carbonate $\delta^{13}C$ compositions are in the higher range for carbonates formed in terrestrial soils (this applies in particular for Jona and Aabach sections). The most plausible driver for elevated pedogenic carbonate $\delta^{13}C$ values of the mid-Miocene North Alpine foreland*

*basin are diminished soil respiration rates, as suggested by the reviewer. We agree with the reviewer that on a very local scale SMB pedogenic carbonate might have been formed during rather dry conditions. Dry conditions with pronounced soil evaporation and associated higher $\delta^{13}C$ values can be found in proximal alluvial fan settings where soil carbonate forms in coarse host rock and soil water retention potential is likely to be diminished (Schwartz, Theresa & Methner, Katharina & Mulch, Andreas & Graham, Stephan & Chamberlain, Charles. (2019). Paleogene topographic and climatic evolution of the Northern Rocky Mountains from integrated sedimentary and isotopic data. Geological Society of America Bulletin. 131. 10.1130/B32068.1.).*

- *However, no paleobotanical and paleozoological evidences are given for a general arid setting with moderate, or even desertic vegetation cover for the Mid-Miocene Swiss Molasse Basin. (Bolliger, T., Neuhausen, H. G. and Hantke, R., Zur Geologie und Paläontologie des Zürcher Oberlandes, Vierteljahrsschrift der Naturforschenden Gesellschaft Zürich, 133(1), 1–24, 1988.) report numerous plant and fossil sites for the OSM of the Hörnli alluvial megafan and reconstruct a humid subtropical climate with rather high precipitation rates interrupted occasionally by drier and/or cooler climate phases. Evidences for warm, humid to tempered climate characterized by subtropical and deciduous forests for Early to Middle Miocene are also found by (Böhme, M., Bruch, A. A. and Selmeier, A.: The reconstruction of Early and Middle Miocene climate and vegetation in Southern Germany as determined from the fossil wood flora, Palaeogeogr. Palaeoclimatol. Palaeoecol., 253(1–2), 107–130, doi:10.1016/j.palaeo.2007.03.035, 2007.) who provides mean annual precipitation rates ranging from ~830–1350 mm.*

- *Given these high MAP estimates, it would be hardly possible to form or preserve any pedogenic carbonate (under modern atmospheric conditions) (Breecker, D. O., Sharp, Z. D. and McFadden, L. D.: Seasonal bias in the formation and stable isotopic composition of pedogenic carbonate in modern soils from central New Mexico, USA, Bull. Geol. Soc. Am., 121(3–4), 630–640, doi:10.1130/B26413.1, 2009, and Zamanian, K., Pustovoytov, K. and Kuzyakov, Y.: Earth-Science Reviews Pedogenic carbonates : Forms and formation processes, 157, 1–17, doi:10.1016/j.earscirev.2016.03.003, 2016.)*

- *This suggests that either the climate in the Northern Alpine Foreland Basin was very seasonal with pronounced dry periods or that the $\rho CO_2$ levels were higher than previously thought. The latter is under investigation and more recent studies (e.g. Sosdian, S. M., Greenop, R., Hain, M. P., Foster, G. L., Pearson, P. N. and Lear, C. H.: Constraining the evolution of Neogene ocean carbonate chemistry using the boron isotope pH proxy, Earth Planet. Sci. Lett., 498, 362–376, doi:10.1016/j.epsl.2018.06.017, 2018.) have proposed that the atmospheric $\rho CO_2$ levels were indeed higher than the first estimates for this time period.*

- *This topic has been touched in the paper of Methner et al. (2020). In this study we prefer to focus on paleoelevation reconstruction based on inferred $\delta^{18}O$ values in precipitation.*

4) This brings me to my chief concern about the paper's conclusion that paleoelevation was ~4200 m. That is: if the setting was that desertic, how can one be confident that evaporation does not influence even the lowest d18O values. I understand that the authors tried to minimize this by using only the lowest 25% quartile. That should help. But were this my paper, I would think this through very carefully, and perhaps be more conservative. Evaporation would expand the difference between isotopic values from low and high elevations and lead to overestimates of paleoelevation. In two papers from 2007, I tried to assess the effects and limits of evaporation on isotopic values from soils in dry climates (there are probably better papers on there on this topic that I am unaware of):

Quade, J., Garzione, C., and Eiler, J., 2007, Paleosol carbonate in paleoelevation reconstruction, *in* M. Kohn, ed., Paleoelevation: Geochemical and Thermodynamic approaches. Reviews in Mineralogy and Geochemistry, Mineralogical Society of America Bulletin, v. 66, p. 53-87.

Quade, J., Rech, J., Latorre, C., Betancourt, J., Gleason, E., Kalin-Arroyo, M., 2007, Soils at the hyperarid margin: the isotopic composition of soil carbonate from the Atacama Desert. *Geochimica et Cosmochima Acta* 71, 3772-3795.

This manuscript cites the first paper but I am not sure they fully internalized the meaning of the results, because mid-elevation (say up to 2000 m) soil carbonate from the Great Basin is the best analog for carbon isotopes in the molasse basin, and those soils show pretty strong (but variable) evaporation effects. In short, I come away with the feeling that there was some evaporation effect even in the lowest 25% quartile of d18O values, and therefore that 4200 m is a maximum estimate.

I don't expect the authors to change the manuscript on this point. They are free to disagree. But I urge them to think more carefully on this point and revise the manuscript if they see fit to do so, or not at all.

- *We acknowledge the concern of the reviewer on this point. We agree with the reviewer that if all obtained $\delta^{18}O$ data (included the lowest 25%) was impacted by evaporation, this would result in inferred paleoelevation estimates biased towards maximum elevation differences.*
- *For the following reasons we assume that Swiss Molasse Basin $\delta^{18}O$ records were not systematically affected by evaporation:*
  1) *Comparison with data from GNIP stations in Switzerland (~250–600 m.a.s.l.) shows, that modern precipitation $\delta^{18}O_w$ compositions reach values of ~ -8‰ – -2‰ for the summer months (June–August), which is in good agreement with our inferred near sea level $\delta^{18}O_w$ value of -5.8‰ for the Swiss Molasse Basin (we included this in the revised manuscript, see lines 381–383). Moreover, considering higher mid-Miocene summer temperatures compared to today, we can assume that precipitation $\delta^{18}O_w$ values may have been more elevated during the mid-Miocene.*
  2) *Reconstructed $\delta^{18}O_w$ values from volcanic ash layers (Bauer et al., 2016) yield values between -6.1 and -2.9‰ and are in consent with our SMB $\delta^{18}O_w$ estimate (see lines 377–381 in revised manuscript).*
  3) *Presumably the moisture in the Northern Alpine Foreland Basin came partly from the Molasse Sea, which was located in closer proximity to collected SMB deposits in the Miocene. Thus, vapour masses which travelled inland had to cover shorter distances between the source water body and their destination area. Consequently, according to the isotopic continental effect, mid-Miocene precipitation $\delta^{18}O_w$ values are not supposed to obtain lower values than modern precipitation $\delta^{18}O_w$ values.*

4) *As replied in the comment to remark 3) we consider a dry, desertic climate very unlikely for the mid-Miocene Swiss Molasse Basin as various paleobotanical studies indicate a temperate to humid climate with moderate to high precipitation rates and (qualitatively) high plant density. However, we cannot completely exclude that individual SMB carbonate samples were affected by (soil) evaporation on a local scale. For this reason, and in order to reduce potential soil evaporation-derived bias, we calculate paleoelevation estimates only with the lowest 25% $\delta^{18}O$ values and consider this as a sufficient precautionary measure.*

5) *Difference from the previous study (Campani et al., 2012) is given by i) a newly measured ambient temperature for mineral-water isotope exchange during soil carbonate formation and ii) a different choice of sea level reference section, which contribute to +1.9‰ and +1.2‰, respectively (see lines 374–376 in revised manuscript). In the case of a systematic evaporation bias (which we do not expect, see arguments above), first, the Aabach section would be the most affected, and second, evaporation would result in +1.2‰ between the Fontannen and Jona sections. Given the modern isotopic lapse rate of -2.0‰/ km (Campani et al., 2012) this results in a $\Delta z(m)$ of "only" +600 m, and cannot be resolved from the calculated propagated error in paleoelevation of ±770m.*

5) I was surprised by the really high soil T°C (47) found by the Methner paper; it will be interesting if this can be reproduced elsewhere for the MMO.

- *We agree with the reviewer that $\Delta_{47}$ soil temperatures for the Swiss Molasse Basin section "Fontannen" as provided by Methner et al. (2020) are rather high, but maybe more interestingly, comprise a large temperature span of ~15°C, also providing low $\Delta_{47}$ soil temperatures within the same section. We find similar high temperatures for the two other sections Jona and Aabach in this study.*
- *Furthermore, in order to exclude technical errors, we tested material from two different carbonate samples on a second mass spectrometer with a different technical setup (coupled with the automated carbonate device KIEL IV) and we obtained comparable temperature within the error range.*
- *As explained in the comment to remark 1) we suggest that sampled carbonate nodules have not been exposed to diagenetic overprint.*
- *We conclude that the rather high $\Delta_{47}$ soil temperatures found for the Swiss Molasse Basin sections is a very interesting finding and we intend to conduct further studies in this context.*

Here are a few line-by-line comments and edits

14: omit , however; omit geochemistry

- *We changed the text according to this suggestion and deleted the terms "however" and "geochemistry".*

16: sea-level here and elsewhere, where used as an adjective

- *We changed the text according to this suggestion and added a hyphen whenever the term sea-level was used as an adjective.*

22: state the range of dD values

- *We specified the phyllosilicate $\delta D$ value which was used for paleoelevation calculation.*

76: molasse

- *We changed the text according to this suggestion.*

96-97: were predominantly composed

- *We changed the text according to this suggestion and replaced "are predominantly composed" with "were predominantly composed".*

103: is the Molasse Sea a formal name?  otherwise no caps

- *The term "Molasse Sea" represents a formal name.*

114: astronomically tuned

- *We changed the text according to this suggestion and replaced "astronomically-tuned" with "astronomically tuned".*

156: What typical depths are the nodules below the top of the paleosol?

- *See comment on remark 2).*

205: VSMOW

- *We added an explanatory sentence in the method chapter 3.2 introducing the term VSMOW for $\delta^{18}O$ values. See also reply to comment on sentence 242.*

235: While mostly true, in some settings soil carbonates can form in the cool season, if summer are wet.  Huntington's group at Washington has documented this.  This scenario should be considered.

- *We clarified this in the manuscript by referring to studies which provide evidence for cool season carbonate formation (see lines 283–284 in revised manuscript).*

242 onward: here and in following paragraphs it is essential to insert VSMOW where referring to d18O values, since in many papers, VPDB is the convention for carbonates. This will clear up any confusion.

- *We clarified this by including an explanatory sentence, stating which reference frame has been used for the isotopic systems (see lines 212–213 in revised manuscript).*

295: yes, I agree here, but to develop +1.2‰ carbon isotope values requires dry conditions (mid-elevations of the Mojave Desert are good analogs), near-surface depths of nodule formation, or high pCO2. The last is not indicated from mid-Miocene records elsewhere, although perhaps the mid-Miocene optimum should be lookd at more carefully. That leaves some combination of the forst two explanations: moderate, desertic vegetation cover and a mix of soils depths 0-100 cm deep). Quade et al., 1989, 2008 (on paleoaltimetry) and Breecker et al, 2009 are the authoritative papers on this.

- *See detailed comment on remark 3).*

300: this covariance is also observed in modern soils (Cerling 1984) and other papers.

- *We included the study of Cerling (1984) and Cerling and Quade (1993).*

305: how do you know the Jona section is the best? Explain. From Fig. 2, the Jona section looks the most variable isotopically, and therefore the most impacted by evaporation.

- *We agree with the reviewer that the Swiss Molasse Basin section Jona shows the highest variability in both, pedogenic carbonate $\delta^{18}O$ and $\delta^{13}C$ values. We give detailed explanations on Jona pedogenic carbonate $\delta^{18}O$ and $\delta^{13}C$ values in the comment on remark 2) and 3), respectively.*

323: which mineral pairs? Clarify how this was done.

- *According to the suggestion of the reviewer we clarified and included in the text, that we used the smectite-water oxygen isotope fractionation factor of Sheppard and Gilg (1996) to recalculate Swiss Molasse Basin $\delta^{18}O_w$ values from the $\delta^{18}O$ values of volcanic ashes.*

328: no new paragraph?

- *We changed the text according to this suggestion and removed the paragraph.*

352: good! Few people know about the Sharp paper, but it is ahead of its time

- *We fully agree with this comment.*

394: million years

- *We changed the text according to this suggestion and replaced "Myr" with the term "million years" as recommended by the reviewer.*

**Citation**: https://doi.org/10.5194/se-2021-59-RC1

---

## Author Comment (AC2)

**Author response:** We would like to thank the referee for the time and effort he dedicated in reviewing our manuscript. We appreciate the referee's insightful comments and suggestions and carefully addressed them. Please see below, in blue and italic font, for a point-by-point response to the reviewer's comments. Provided page numbers refer to the revised manuscript file with tracked changes.

**RC2**: 'Comment on se-2021-59', Peter van der Beek, 30 Jun 2021 reply

Krsnik et al. present new stable and clumped oxygen / carbon isotope data from three early – middle Miocene sections in the North Alpine foreland basin, which they combine with existing data from a high-elevation site (Simplon Fault Zone) and isotope-enabled climate models to refine earlier estimates of middle-Miocene paleo-elevation of the Swiss central Alps. They find that this paleo-elevation was probably significantly higher than the present-day, confirming early data that were generally considered with some skepticism. This is a good paper with interesting new data that will make a nice contribution to *Solid Earth*. I would recommend acceptance after moderate revision as there are a few aspects that could be made clearer:

1) First, the manuscript does not make it entirely clear what is new and what is existing data. It appears that the oxygen- and carbon-isotope data from the three sections were collected specifically for this paper, while the hydrogen-isotope data from the Simplon Fault Zone are from Campani et al. (2012). However, what about the clumped-isotope data? Some of these appear to be from Methner et al. (2020). Was additional data collected for this manuscript? A data table that explicitly states the origin of the data would be useful.

- *Following the suggestion of the reviewer we prepared a table clearly displaying which data was obtained specifically for this study and which data was provided by Campani et al. (2012) and Methner et al. (2020). This table can be found in the Supplementary data (SI6).*
- *We also made sure to provide a more precise referencing to these studies.*

2) Similarly, it is not clear whether the paleoclimate simulations were run specifically for this study or whether they were taken from Botsyun et al. (2020). It is totally OK to reuse data or models but their origin should be clear.

- *Paleoclimate simulations from Botsyun et al. (2020) were not run specifically for this study. We state the origin of these climate simulations in lines 262–263 in the manuscript.*

3) Second, I feel that the sections, data and time constraints could be described a bit more clearly. In particular, Fig. 3 (which should be Fig. 2 – see below) does not contain a lot of information: it would be good if this figure showed stratigraphic names, specific age markers discussed in the text (with their age), the tie to the paleomagnetic time scale, etc. Carbon-isotope data are discussed but not shown at all; these could be plotted in the panels of Fig. 3 adjacent to the oxygen data. Similarly, it would be useful to show the locations of the samples collected for clumped-isotope analysis on the logs and report the inferred paleo-temperatures in the figure.

- *We appreciate this comment and implemented all of the suggestions of the reviewer as it really improves the figure.*
- *Following the suggestions we changed the order of Fig. 2 and Fig. 3.*
- *We added more information to the (new) Fig. 2 including bentonite horizons and mammal sites with their ages, respectively mammal zones, and locations of samples for clumped isotope analysis with corresponding ($\Delta_{47}$) temperatures. We added two additional columns displaying the regional magnetostratigraphy used for calculation of soil carbonate ages and showing stratigraphic names of the OSM sediments, respectively. Furthermore, we included $\delta^{13}C$ records for all three Swiss Molasse Basin sections.*
- *Additionally, and not related to the suggestions of the reviewer, we changed the order of displayed Swiss Molasse Basin records from "Fontannen-Jona-Aabach" to "Fontannen-Aabach-Jona" according to their geographical locations for the purpose of better readability.*
- *In addition to Fig. 2 in the manuscript we prepared a more detailed figure for the Supplementary Material displaying magnetostratigraphic logs and their correlation to the paleomagnetic time scale for each section (Fig. SI1 in suppl. Material SI5).*

4) I would also like to see a somewhat more complete description of the paleoclimate models: what is the "ECHAM5-wiso GCM"? I don't think one can assume the average reader of *Solid Earth* to be acquainted with these acronyms. What is meant by a "pre-industrial model setup"? Does this only apply to the paleogeographic or also to the climatic (i.e., atmospheric $pCO_2$) boundary conditions? If pre-industrial $pCO_2$ was used instead of an estimated middle-Miocene condition, what would be the influence on the model predictions? Would they be realistic? Could a "distant region" for which middle-Miocene stable-isotope data are available be included and used to calibrate the model? Overall, this model description section needs a bit more explanation and justification.

- *As suggested by the reviewer we complemented section 3.4 ("Paleoclimate simulations") by adding relevant information about the climate model ECHAM5-wiso. This comprises details on the origin of the ECHAM5-wiso, resolution of the model setup, and boundary conditions of the pre-industrial model setup.*
- *It is beyond the scope of this contribution to conduct further experiments with Miocene boundary conditions. Current efforts by the original author (S. Botsyun) aim at complementing climate model runs with Miocene boundary conditions. This, together with model validation against proxy data at different locations (including "distant regions"), is part of a future project.*
  .

5) I suppose the paleoclimate models make predictions of the (summer – JJA) temperatures at the fan sampling site**.** It would be interesting to report these and compare them to the estimates obtained from the clumped-isotope analysis; on the one hand to provide independent support for these fairly elevated temperature estimates and on the other hand to calibrate / assess the model outcomes.

- *Clumped isotope analyses provide temperatures prevalent during the time of soil carbonate formation, which is an essential factor to calculate mineral-water isotope fractionation. We, therefore, use ($\Delta_{47}$) temperatures of soil carbonates for calculation of $\delta^{18}O$ in Swiss Molasse Basin soil waters and ultimately precipitation. We agree that*

*obtained Swiss Molasse Basin ($\Delta_{47}$) temperatures appear rather high and we acknowledge that the role of $\Delta_{47}$ based soil temperatures as a proxy for air temperatures is currently being debated. Nevertheless, we want to highlight that we use the obtained ($\Delta_{47}$) temperatures solely as a means for reconstruction of the mineral-water fractionation temperature of the formed soil carbonate. For the following reasons we do not expect a close match between the temperatures simulate in (Botsyun, S., Ehlers, T. A., Mutz, S. G., Methner, K., Krsnik, E., & Mulch, A. (2020). Opportunities and challenges for paleoaltimetry in "small" orogens: Insights from the European Alps. Geophysical Research Letters, 47, e2019GL086046. https://doi.org/10.1029/2019GL086046) and the ($\Delta_{47}$) temperatures presented here:*

- *1) In the absence of a Miocene model setup, climate model simulated summer mean (June-July-August) temperatures for the Northern Alpine Basin were obtained with pre-industrial boundary conditions and yielded ~16°C (Supplementary Material of Botsyun et al., 2020, Fig. S4). The pre-industrial $\rho CO_2$ (280 ppm) boundary conditions restrict the upper temperature limit and therefore the simulated temperatures are lower than one would expect for the Middle Miocene. Because of the pre-industrial model setup, simulated temperatures and measured ($\Delta_{47}$) temperatures of Middle Miocene soil carbonates record different climatic conditions and, in our case, are not directly comparable. As shown in the study of Methner et al. (2020), temperatures changed dramatically towards lower temperatures during the Middle Miocene Climate Transition when compared to the Middle Miocene Climate Optimum covered here. The current model setup is not able to resolve such temporal differences.*

- *2) A comparison between modeled and measured ($\Delta_{47}$) temperatures would require consideration of the 2-3 times higher $\rho CO_2$ for the Miocene. We therefore expect simulated Middle Miocene temperatures to be up to 6–12°C higher, if climate sensitivity (temperature increase in response to a doubling of $pCO_2$) of 1.5-6°C is taken into account. This would reduce the difference between modeled and measured ($\Delta_{47}$) temperatures for the Swiss Molasse Basin.*

6) When discussing the results and their implications, a fuller assessment of uncertainties could be made, in particular considering the uncertainties in lapse rate. Why not first give the full range of possible paleo-elevations considering the different lapse-rate models and then potentially discuss a preferred option?

- *We very much appreciate this comment and followed the suggestion of the reviewer.*
- *Besides calculating paleoelevation based on the isotope lapse rate of -2.0‰/ km ±0.04‰ (Campani et al., 2012), we now discuss the impact of a more conservative choice of isotope lapse rates (taken from Botsyun et al., 2020) and provide maximum $\Delta z$ (m) and minimum $\Delta z$ (m) based on the uncertainty of $\Delta(\delta^{18}O_w)$ for both lapse rates (lines 409-411 in revised manuscript).*
- *In doing so, we realized that a copy-paste mistake was present in the original manuscript whenever referring to the uncertainty of $\Delta(\delta^{18}O_w)$, which was falsely given with "±0.5‰". We now give the correct error of "± 1.5‰" (which was already reported in Table T5 in Supplementary Material SI7, and correctly used to calculate elevation uncertainties. We now also present this error in panel b) of Fig. 6.*
- *As elevation uncertainties were calculated with the correct parameter, this mistake has no implications for the paleoelevation calculations, therefore no revision of the stated paleoelevation is needed.*

7) Also, it seems that the lapse rate predicted by the paleoclimate model is significantly higher than the observed modern lapse rates, whereas it is argued in lines 348-349 that the mid-Miocene lapse rates should probably be lower than the modern. Why is this – is it linked to the climatic boundary conditions used in the model (see above)?

- *This is correct. Please also see our response to remark 5): The model from Botsyun et al (2020) does not reflect Miocene boundary conditions. We would expect lower lapse rate value for a warm (and more humid) atmosphere as has e.g. been suggested by Poulsen, C. J. and Jeffery, M. L.: Climate change imprinting on stable isotopic compositions of high-elevation meteoric water cloaks past surface elevations of major orogens, Geology, 39(6), 595–598, 2011.*

8) Finally, while the presentation of the results and their interpretation in terms of paleo-elevation is fairly rigorous (as far as I can judge), the final part of the discussion (section 5.5) suddenly becomes quite vague, arm-wavy and speculative. For instance, it is unclear if the authors are arguing for high elevation in the Lepontine dome or in the Aar massif at 14 Ma. It is important to clarify the spatial scale to which the paleo-elevation estimate pertains – and would this number constrain the average or the maximum elevation in this region? I feel this discussion could be improved by integrating the drainage development as constrained by provenance data. As long as there was a direct connection between the Lepontine dome and the studied fans in the foreland basin, the Aar massif could not have been elevated – this is a very important piece information that should be better integrated in the scenario. It has been argued in the French western Alps that the Internal Zone (southeast of the Penninic Front) was elevated substantially earlier than the External Crystalline Massifs (e.g., Fauquette et al., *Earth Planet. Sci. Lett.* 2015); a similar scenario appears to apply to the central Alps from the present data. Making such linkages would help developing a more holistic view of Alpine paleotopography.

- *We relate this criticism to the rather confusing way of how we have structured this part of the discussion. We wanted to make the point that (i) the relatively high elevation for the area surrounding the western margin of the Lepontine Dome (inferred from our data) contrasts to evidence for a low-elevation topography in the area of the Lepontine Dome itself (such as a low sediment supply to the basin following tectonic unroofing), and that (ii) the time with these inferred elevation contrasts coincides with the period when the reorganization of the drainage network started. As a consequence, while the Central Alpine landscape was most likely cylindrical between the Late Oligocene to Early Miocene and was most likely characterized by a regular spacing and a constant relief between the valleys, the post 20 Ma Central Alpine landscape became more complex and most likely non-cylindrical.*
- *We revised section 5.5 ("High (and highly variable) mid-Miocene Central Alps?"; lines 433–446 in revised manuscript) and section 6 ("Conclusions"; lines 508–524) to provide more clarity.*

9) Apart from these main issues, I have a number of more minor editorial comments, which are listed below tied to line numbers. Overall, the manuscript is well written and easy to read. A few references are missing from the reference list and a more generous use of commas could be made.

1 (Title): whereas the manuscript discusses the mid-Miocene paleo-elevation of the Central Alps, there is little discussion of paleo-relief. I would suggest that this is either added more prominently to the discussion (if the data allow constraining some measure of paleo-relief) or the title is modified.

- *We changed the title according to the reviewer's suggestion and deleted "and high relief" from the title.*

22: the acronym SFZ has not been explained at this stage. In general, please try to minimise the use of acronyms as they detract from the reading in exchange for only a limited gain in space.

- *We changed the text according to this suggestion and wrote out the term Simplon Fault Zone.*

36-38: this phrase ("The European Alps are …") seems somewhat out of place here and should be moved or modified / expanded.

- *According to this suggestion we moved this phrase to lines 28–29 in the revised manuscript.*

39-43: this paragraph could benefit from being a bit more specific. Where were the cited paleo-elevation estimates obtained, based on what methods? Also, Kocsis et al. (2007) seems to be missing from the reference list.

- *According to this suggestion we revised this paragraph and added more details (see lines 47–58 in the revised manuscript).*
- *We added Kocsis et al. (2007) to the reference list.*

43-45: this appears a bit like setting up a strawman argument; Hergarten et al. (2010) is a very problematic study that is stained by serious flaws in the reasoning. I do not think this is needed or even appropriate as a justification for the current study.

- *According to the suggestion of the reviewer we removed this phrase and the associated reference.*

69: Handy et al. (2010) appears to be missing from the reference list.

- *We added Handy et al. (2010) to the reference list.*

73: SMB, NAFB – see previous comment regarding acronyms; I don't think these are useful here.

- *We followed the suggestion of the reviewer and reduced the use of the acronyms SMB, NAFB, and SFZ.*

89: it would be useful to add a discussion of the evolution of drainage patterns and the implications for (surface) uplift of the Aar massif to this paragraph, as these will aid in sharpening the discussion in section 5.5.

- *See reply to remark 8).*

118: Fig. 3 is called before Fig. 2 and it would be logical to change the order of these figures.

- *We changed the order of Fig. 2 and Fig. 3.*

135-138: it would be useful to show the stratigraphic levels of the dated bentonites as well as the mammal sites (with their corresponding mammal zone) on the logs.

- *We followed the suggestion of the reviewer and added stratigraphic positions of the bentonites (with their dated ages) and mammal sites (with corresponding zones) to Fig. 2. Furthermore, we added sampling sites of the samples used for carbonate clumped isotope measurements and the magnetostratigraphy used for sample age calculation (see reply above).*

Also, a line of explanation about how a conglomerate in one section can be correlated to a limestone in the other would be welcome.

- *We changed the text according to this suggestion and added a brief explanation about the correlation between the Hüllistein conglomerate and the Meilen Limestone (see lines 164–167 in revised manuscript).*

150-151: see above comment. Also, were magnetostratigraphic analyses performed on these sections? If so, why not show the magnetostratigraphy as well? The age constraints are important here so it would be good to clearly show these constraints on the figure.

- *We changed Fig. 2 according to this suggestion and added the magnetostratigraphy.*

160: "magnetostratigraphy" rather than "paleomagnetostratigraphy". Also, this was not discussed in section 2, but should have been if such data are available (see above comment).

- *We replaced "paleomagnetostratigraphy" by "magnetostratigraphy" according to this suggestion. Furthermore, we added a brief discussion about the magnetostratigraphic constraints to each of the three sections (see lines 150–153; 167–168; 181–182 in revised manuscript).*

181-183: these two sentences would read a bit more easily if the starting subphrase was moved to the end of the main phrase (e.g., "Ascending air masses undergo adiabatic cooling and rain out with increasing altitude, which leads to …"; and similarly for the following phrase).

- *We rephrased the sentences according to this suggestion (see lines 217–219 in revised manuscript).*

183: add "altitudinal" to "lapse rates" for clarity.

- *We added "altitudinal" according to this suggestion.*

225: see major comment on description of climate simulations.

- *We appreciate this comment very much and give a detailed reply within the major comments. See comment on remark 4).*

230: it is not clear what the "(250 m)" pertains to.

- *In the Botsyun et al (2020) paper the "0 % Alps" topography has the Alps set at 250 m above sea level; hence the "(250 m)" term. We tried to clarify and added "topography set to 250m" (see lines 275–276 in revised manuscript).*

233-234: "enhances assessment of paleoclimate changes" is quite vague – can you elaborate? Is there any data available for such a distant location that could help constraining the model?

- *See comment on remark 5).*

240-242: this phrase doesn't read very well; maybe try a construction with "Although …"?

- *According to this suggestion we rephrased this sentence to improve readability (see lines 290–291 in revised manuscript).*

248-251: is this new or existing data? Can it be shown on the log and/or a separate data figure?

- *$\delta^{18}O$ and $\delta^{13}C$ data for the Fontannen section has been provided by Campani et al. (2012). $\Delta_{47}$ based temperatures are from Methner et al. (2020).*
- *We added the origin of the data in the text (see lines 297–298 in revised manuscript) and the figure caption of Fig. 2.*

287: why would the proximal part of the fan be at "more than" $300 \pm 100$ m above sea level, when it appears that the uncertainties have already been included in this calculation?

- *We changed the text according to this suggestion and deleted "more than".*

290: "(mainly also because of the occurrence of paleolakes …)" is a bit of a mysterious addition to this phrase – either explain this or remove it.

- *We changed the text according to this suggestion and removed the sub-clause "mainly also because of the occurrence of paleolakes ..." (see lines 348–349 in revised manuscript).*

293 (and 305): Fig. SI3 could easily be made part of the main paper, which is not very long in any case. Having this figure in the main paper would facilitate assessing this argument.

- *We appreciate this comment and acknowledge the proposal of the reviewer. The aim of our study is to provide paleoelevation calculations for the mid-Miocene Central Alps, which are based on measuring $\Delta(\delta^{18}O)$ between two sites. Pedogenic carbonate $\delta^{13}C$ ratios are not an essential part for these calculations and it is beyond the scope of this study to examine in detail the significance of Swiss Molasse Basin carbon isotope compositions which are driven by complex processes within the soil.*
- *Therefore, rather than including a $\delta^{13}C/\delta^{18}O$ cross plot in the manuscript, we prefer to add the $\delta^{13}C$ data to Fig. 2 (as suggested by the reviewer in remark 3), and furthermore*

*provide Swiss Molasse Basin pedogenic carbonate $\delta^{13}C$ values in a separate data table in the Supplementary Material (SI7). We give the $\delta^{13}C/\delta^{18}O$ cross plot in the Supplementary Material (SI5).*

305-306: argument c) has not been developed previously and it is thus not clear why using this section location would underestimate paleo-elevations. Please provide an explanation.

- *We changed the sentence and included an explanation according to this suggestion (see lines 365–366 in revised manuscript).*

322-325: the oxygen-isotope data from volcanic ash horizons could be plotted in Fig. 6 for simpler comparison with the data presented here.

- *According to the suggestion of the reviewer we plotted $\delta^{18}O_w$ values derived from volcanic ashes in the Fig. 6a.*

330: Equation (1) appears pretty obvious; it is not clear why this equation is given and not others that are maybe less straightforward (e.g., for the isotopic fractionation of the lapse rates).

- *According to the suggestion of the reviewer we deleted the equation since it is not essential for understanding the paleoelevation calculation (see lines 393–395 in revised manuscript).*

331-353: see major comments on assessment of uncertainties and the model-predicted lapse rate above; these could be discussed here.

- *We appreciate this comment and revised the text according to this suggestion. See comment on remark 6).*

350: please provide the present-day (average or peak) elevation of the relevant area for direct comparison with this number.

- *We added the elevation of the neighbouring peak (Monte Leone with 3553 m.a.s.l.) according to the suggestion of the reviewer (see line 422 in revised manuscript).*

357-359: the comparison between Figs. 5b and 6b is not straightforward and I am wondering whether there would be a more efficient way of showing the model – data comparison?

- *According to the suggestion of the reviewer we revised Fig. 6 and included an additional panel showing climate modeled Swiss Molasse Basin $d^{18}O$ data for the case Alps150.*

364-370: a fairly big interpretational step seems to have been taken here. This section could be rewritten to take a more linear course from the paleoelevation estimates to implications for paleo-topography in the Alps to potential geodynamic implications.

- *We revised section 5.5 ("High (and highly variable) mid-Miocene Central Alps") and moved this paragraph to the introduction (see lines 31–37 in revised manuscript).*

389-393: see major comment on drainage development above: when was the connection between the Lepontine dome and the fans cut off by surface uplift of the Aar massif? This is an important constraint on the evolution of topography. By the way, Bernard et al. (in press) has now been published.

- *See comment on remark 8)*
- *We revised the reference and included the year of publishing.*

398-400: OK here is some of that discussion – this should just be made clearer and stated more upfront.

- *We revised this section (see lines 433–446 and 454–464 in revised manuscript).*

403: whether mean elevation increased or decreased related to extensional denudation of the Lepontine dome footwall would depend on the considered scale: some of the metamorphic core complexes in the western USA stand up to 2 km above their surroundings. The spatial resolution of the paleo-elevation estimate is key here

- *We completely agree with the reviewer here. The elevation of a region undergoing extensional denudation will (amongst other parameters) depend on the rate at which temperature anomalies in the exhumed footwall are being relaxed. Ultimately, the end result of extensional detachment faulting and thinning of (buoyant) continental crust should be a lowering of elevation compared to the pre-extensional stage. It is hence tricky to infer relative elevation differences between neighboring regions undergoing differential amounts of extensional deformation. Given the rather high elevations obtained here and the absence of evidence for low-δD meteoric fluids in mylonites further East (e.g. the Brenner fault zone; see Table 3 in "Mancktelow, N., Zwingmann, H., Campani, M., Fügenschuh, B. and Mulch, A.: Timing and conditions of brittle faulting on the Silltal-Brenner Fault Zone, Eastern Alps (Austria), Swiss J. Geosci., 108(2–3), 305–326, doi:10.1007/s00015-015-0179-y, 2015.") led us to suggest that the overall effect of extensional faulting may have been represented by lower elevations when compared to the Simplon region.*
- *See also reply to comment on line 406.*

406: it is not clear what evidence was provided for "co-existence of regions with different elevations on a small spatial scale …".

- *Our conclusion is based on the estimated paleoelevation (as inferred from our data) of >4000 m for the region surrounding the Simplon Fault Zone (SFZ), which is in close proximity (~45 km to the W) to the Lepontine Dome. For the latter decreased sediment discharge rates were suggested for the same time interval (Kuhlemann, J., Frisch, W., Dunkl, I. and Székely, B.: Quantifying tectonic versus erosive denudation by the sediment budget: The miocene core complexes of the Alps, Tectonophysics, 330(1–2), 1–23, doi:10.1016/S0040-1951(00)00209-2, 2001.). Consequently, we link the decreased sediment fluxes with a low-elevation topography in the area of the Lepontine Dome when compared to >4000 m inferred for the SFZ.*

410-412: again, the key question is the spatial scale on which the paleo-elevation estimate constrains paleo-topography, and what aspect of the topography (mean, maximum?) is actually constrained.

- *δ-δ paleoaltimetry provides constraints on the mean elevation of the catchment of precipitation which falls in the area of the high-elevation site. Within the catchment, runoff collects at the lowest topographic point, and the measured $\delta^{18}O_w$ estimate represents an integrated signal originating from different elevations within this area. In our case, the inferred paleoelevation estimate represents the mean elevation of the mid-Miocene paleo-catchment of the Simplon area.*
- *We clarified this in the text (see lines 405–407 in the revised manuscript).*

426: is the evidence for *uplift* or *exhumation* of the Aar massif at ~20 Ma?

- *We replaced the term "uplift" with "exhumation".*

429-430: it would be helpful to place this number into perspective by quoting the relevant present-day elevation measure.

- *According to the suggestion of the reviewer we included the present-day elevation of this section (see line 509 in the revised manuscript).*

Fig. 2 should become Fig. 3. Labelling each photo individually (a – h) would help identifying the panels.

- *We changed the order of Fig. 2 and Fig. 3 and labelled each photo individually as suggested by the reviewer.*

Fig. 3 should become Fig. 2. This figure should include the chronostratigraphic constraints (age markers, magnetostratigraphy if existent) and stratigraphic names (mentioned in the text). It would be helpful to add the carbon isotope data (using a double scale and a slightly different colour or symbol) as well as at least the locations of the samples used for clumped-isotope analysis.

- *See comment on remark 3). We followed the suggestion of the reviewer and added age markers, magnetostratigraphy, stratigraphic names, and locations of samples used for carbonate clumped isotope analysis.*

Fig. 6: panel (b) needs a legend for the different lapse rates. The green bar indicating the paleo-elevation estimate should take into account the uncertainties in both $\Delta(\delta^{18}O_w)$ and in the lapse rates.

- *We appreciate this comment and revised panel b) in Fig. 6 according to the suggestions of the reviewer as we think that this improves the figure very much.*
- *We very much appreciate the argument regarding the uncertainty in $\Delta(\delta^{18}O_w)$ since this is an essential factor when calculating paleoelevation. Therefore, we give inferred paleoelevations including the full error span in $\Delta(\delta^{18}O_w)$ based on all four isotope lapse rates (Table SI5 in Supplementary Material). For the preferred Swiss Molasse Basin section Jona the uncertainty in $\Delta(\delta^{18}O_w)$ of ±1.50 ‰ (propagated error in stable and clumped isotope analysis) and error of the chosen isotope lapse rate result in a total*

*error of ±1.54‰ which equals ±770 m if choosing the lapse rate after Campani et al. (2012).*

- *For the purpose of improved visualization we omit the graphic representation of inferred paleoelevations based on all four isotope lapse rates, and show only calculated maximum Δz (m) and minimum Δz (m) according to the preferred lapse rate after Campani et al. (2012).*
- *Changes in Fig. 6:*
  - *We added a green vertical bar indicating the uncertainty in $\Delta(\delta^{18}O_w)$ and black horizontal lines clearly displaying the calculated maximum Δz (m) and minimum Δz (m) according to the error in $\Delta(\delta^{18}O_w)$.*
  - *We included a legend to panel b) presenting the different isotope lapse rates used for paleoelevation calculation.*

**Citation**: https://doi.org/10.5194/se-2021-59-RC2